# MEGA: Message Passing Neural Networks for Multigraphs with Edge Attributes

## Abstract

Many real-world graphs, such as financial transaction networks, are edge-attributed multigraphs that feature multiple edges between the same pair of nodes, each with distinct edge attributes. State-of-the-art neural network solutions operating on such edge-attributed multigraphs either preprocess the multigraph by collapsing its multi-edges into a single edge or introduce auxiliary edge features that compromise permutation equivariance. We introduce MEGA-GNN, a graph neural network (GNN) for edge-attributed multigraphs, which overcomes these limitations by employing a two-stage aggregation process in its message passing layers: first, features of the multi-edges between the same two nodes are aggregated, and then messages from distinct neighbors are combined. We show that MEGA-GNN computes a richer set of statistical features than the GNNs that implement only single-stage aggregation in their message passing layers. We evaluate MEGA-GNN on seven financial transaction network datasets and three temporal user-item interaction datasets, demonstrating significant improvements in minority-class F1 scores for illicit transaction detection and ROC-AUC scores for user state-change prediction, respectively, compared to state-of-the-art methods.

## 1 Introduction

A multigraph is a graph that allows multiple edges, *multi-edges*, between the same pair of nodes. Multigraphs naturally arise in domains such as transportation, cybersecurity and finance, where repeated interactions between entities are common. Financial transaction networks, in particular, capture the flow of money between entities such as individuals or companies and have emerged as a primary application area of edge-attributed multigraphs, where nodes represent accounts, and edges represent transactions carrying rich numerical and categorical information between the accounts.

There has been a growing interest in applying Graph Neural Networks (GNNs) to financial crime analysis (Hiroki Kanezashi & Hirofuchi, 2022; Cardoso et al., 2022; Nicholls et al., 2021; Weber et al., 2019; Egressy et al., 2024; Lin et al., 2024), driven by the impressive success of GNNs in diverse domains, including biology (Xu et al., 2019; Gilmer et al., 2017), social networks (Veličković et al., 2018; Corso et al., 2020; Hamilton et al., 2017), and knowledge bases (Schlichtkrull et al., 2018; Vashishth et al., 2020; Chen et al., 2021b). Nevertheless, financial crime detection remains a challenging task due to a scarcity of datasets with labeled examples, extreme class imbalance in available datasets, with only a tiny fraction of transactions being illicit, and constantly evolving crime patterns resulting from increasingly complex customer interactions.

The directed and edge-attributed multigraph structure of financial transaction networks presents additional challenges. The limitations of popular GNNs (Xu et al., 2019; Gilmer et al., 2017; Corso et al., 2020; Schlichtkrull et al., 2018) on edge-attributed multigraphs have been clearly demonstrated by Egressy et al. (2024). Multi-GNN (Egressy et al., 2024) is a message passing GNN designed specifically to address the challenges posed by directed multigraphs with edge attributes. It accurately detects complex financial crime patterns in large transaction networks without any feature engineering. Notably, Multi-GNN achieves state-of-the-art performance on both anti-money-laundering (Altman et al., 2023) and phishing-detection tasks (Chen et al., 2021a), significantly outperforming IBM's Graph Feature Preprocessor used in combination with tree-based classifiers (Blanuša et al., 2024). However, the *multigraph port numbering* adaptation of Multi-GNN compromises permutation equivariance because it uses pre-computed port numbers as edge features.

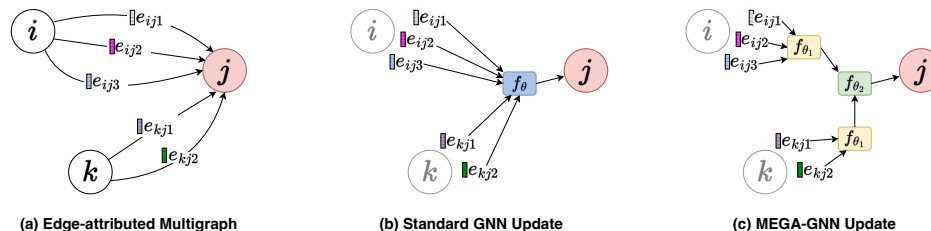

(a) Edge-attributed Multigraph    (b) Standard GNN Update    (c) MEGA-GNN Update

Figure 1: MEGA-GNN performs two-stage aggregation: first, features of multi-edges between the same node pair are aggregated; second, the resulting messages from distinct neighbors are aggregated at the node level. In contrast, standard GNNs (center) perform single-stage aggregation.

ADAMM (Sotiropoulos et al., 2023) is another GNN solution for multigraphs, which aggregates multi-edges into a single edge before message passing, thereby preserving permutation equivariance. However, the message passing layers can no longer utilize the original multigraph and edge features, which limits the effectiveness of ADAMM and its applicability in edge-classification tasks.

While multi-relational GNNs (Schlichtkrull et al., 2018; Vashishth et al., 2020) are highly effective on labeled multigraphs, where each edge corresponds to a well-defined relation type, they are less suitable for financial transaction networks and other edge-attributed multigraphs, where such relations are not explicitly defined. These models apply distinct transformations based on edge types, but lack mechanisms to differentiate between multi-edges (i.e., multiple edges between the same pair of nodes) and edges originating from distinct neighbors. See Appendix A.1 for further details.

To address the aforementioned shortcomings of existing methods, we propose MEGA-GNN, a novel message passing framework designed for edge-attributed multigraphs. MEGA-GNN introduces a **two-stage aggregation mechanism** in its message passing layers: first, attributes of the multi-edges originating from the same neighbor are aggregated; second, the aggregated messages from distinct neighbors are aggregated at the node level. Figure 1 illustrates MEGA-GNN's two-stage message aggregation scheme. Importantly, MEGA-GNN also supports bi-directional message passing, distinguishing incoming and outgoing edges without treating them as undirected. In addition, MEGA-GNN is permutation equivariant because it does not rely on any precomputed features.

The **main contributions** of our work are as follows:

1. We introduce MEGA-GNN, a permutation-equivariant GNN architecture for edge-attributed multigraphs, which performs a two-stage message aggregation based on the given multigraph topology without eliminating multi-edges or using precomputed port identifiers.

2. We prove that MEGA-GNN is more powerful than standard message passing GNNs on edge-attributed multigraphs because its two-stage message aggregation mechanism can compute per-neighbor statistics that cannot be captured by a single-stage aggregation.

3. We show that MEGA-GNN integrates seamlessly with diverse GNN baselines, including GIN (Xu et al., 2019), PNA (Corso et al., 2020), GenAgg (Kortvelesy et al., 2023), and R-GCN (Schlichtkrull et al., 2018), consistently enhancing their performance while maintaining the same asymptotic complexity as baseline GNNs augmented with edge updates.

4. We show that MEGA-GNN establishes a new state of the art across multiple tasks. On anti-money laundering tasks, MEGA-GNN outperforms Multi-GNN Egressy et al. (2024) by up to 10.9% and FraudGT Lin et al. (2024) by 4.8% in minority-class F1 score. For phishing detection on Ethereum blockchain data, it surpasses FraudGT by 7.4% and ADAMM (Sotiropoulos et al., 2023) by over 20%. In user state-change prediction over temporal networks, MEGA-GNN achieves up to 9% improvement over JODIE (Kumar et al., 2019).

## 2 RELATED WORK

The expressive power of GNNs is essential for evaluating their capabilities and has been widely studied (Xu et al., 2019; Barceló et al., 2021; Morris et al., 2023; Bevilacqua et al., 2024; Frasca et al., 2022). Xu et al. (2019) demonstrated that the standard message passing GNNs are limited by the 1-WL (Weisfeiler-Lehman) test for distinguishing isomorphic graphs, and introduced the Graph

Isomorphism Network (GIN), which is provably the most expressive among standard message passing GNNs. Several works have proposed models with expressive power beyond 1-WL, including k-WL-based GNNs (Maron et al., 2019; Morris et al., 2019), randomized node initializations (Abboud et al., 2021; Sato et al., 2021), and particle-filtering-based approaches like PF-GNN (Dupty et al., 2022). The expressiveness of GNNs is closely linked to their universality, as more expressive models can approximate a larger class of graph functions. Sato et al. (2021) showed that unique node identifiers can make GNNs universal but at the cost of losing permutation invariance. On the other hand, Abboud et al. (2021); Loukas (2020) demonstrated that partially randomized node features can achieve universality while preserving permutation invariance. Set functions, like those in Zaheer et al. (2017), have also been shown to be universal under certain input constraints, and Fuchs* & Veličković* (2023) made the connection between a universal set and graph functions. Corso et al. (2020) introduced the PNA model, which improves empirical performance by combining diverse aggregation functions. In Kortvelesy et al. (2023), a learnable aggregation function (GenAgg) is proposed, which improves sample efficiency compared to PNA- and DeepSet-based aggregations in multiset neighborhoods, and can learn to approximate all standard aggregators.

Research on GNNs addressing the challenges of multigraphs, particularly multi-edges with edge attributes, is still in its early stages. ADAMM (Sotiropoulos et al., 2023) proposes collapsing multi-edges into a single super-edge using DeepSet-based aggregation (Zaheer et al., 2017) prior to message passing layers. Although collapsing followed by a message passing layer results in a two-stage aggregation, it occurs only in the first layer. Subsequent message passing layers operate on a modified graph with collapsed multi-edges. Multi-GNN (Egressy et al., 2024) introduces three adaptations: reverse message passing, port numbering, and ego IDs, which together transform baseline GNNs to provably powerful multigraph neural networks. DIAM (Ding et al., 2024) models cryptocurrency transaction networks as directed, edge-attributed multigraphs and focuses on learning node representations that capture temporal transaction patterns (see Appendix A.2 for details).

Multi-level message aggregation schemes have also been proposed in other settings that are not directly applicable to multigraphs. For instance, Hypergraph GNNs (Feng et al., 2019; Huang & Yang, 2021) first aggregate node features within each hyperedge to compute latent hyperedge features, which are then used to aggregate node features. Similarly, P-GNN (You et al., 2019) computes position-aware node embeddings via multi-level message aggregation, first within anchor-sets and then across all anchor-sets. In contrast, our MEGA-GNN architecture directly handles multi-edges, representing multiple different connections between the same pair of nodes.

## 3 MULTIGRAPH MESSAGE PASSING WITH TWO-STAGE AGGREGATION

This section introduces our notation as well as our two-stage aggregation scheme. We theoretically prove that the two-stage approach is strictly more powerful than single stage approaches on multigraphs with edge attributes. Next, we present our MEGA-GNN architecture, which integrates two-stage aggregation into its message passing layers. We further enhance MEGA-GNN with bidirectional message passing for directed multigraphs. Finally, we provide additional theoretical properties of our method, such as permutation equivariance, universality, and inference complexity.

### 3.1 NOTATION

**Definition 1** (Multiset). *A multiset is a 2-tuple $X = (S, m)$ where $S$ is the underlying set of $X$ formed from its distinct elements, and $m : S \to \mathbb{N}_{\geq 1}$ gives the multiplicity of the elements.*

**Definition 2** (Multiset Sum $\uplus$). *Let $A = (S_A, m_A)$ and $B = (S_B, m_B)$ be multisets over a common universe $U$. Their sum $A \uplus B$ is the multiset $C = (S_C, m_C)$ defined by*

$$S_C = S_A \cup S_B, \qquad m_C(x) = m_A(x) + m_B(x) \quad \text{for all } x \in U.$$

*Here, the operator $+$ denotes standard integer addition of multiplicities.*

We denote multisets with $\{\!\{\cdot\}\!\}$ and sets with $\{\cdot\}$. $[n]$ stands for the set $\{1, 2, \ldots, n\}$ for $n \in \mathbb{N}$. Let $\mathcal{G} = (\mathcal{V}, \mathcal{E})$ be a directed multigraph with node set $\mathcal{V}$ and edge multiset $\mathcal{E} = \{\!\{(i, j) \mid i, j \in \mathcal{V}\}\!\}$, where each $(i, j)$ represents a directed edge from node $i$ to node $j$. Let $\mathcal{E}^{\text{supp}} \subseteq \mathcal{E}$ denote the support set of $\mathcal{E}$. We define the edge multiplicity $P_{ij} := m_{\mathcal{E}}(i, j)$, i.e. the number of edges from node $i$ to node $j$. For a node $j \in \mathcal{V}$, the incoming and outgoing neighbors are defined as $N_{\text{in}}(j) = \{i \in \mathcal{V} \mid$

$(i, j) \in \mathcal{E}^{\mathrm{supp}}\}$ and $N_{\mathrm{out}}(j) = \{i \in \mathcal{V} \mid (j, i) \in \mathcal{E}^{\mathrm{supp}}\}$. We consider attributed multigraphs with feature dimensions $d_n, d_e, d \in \mathbb{N}$. Each node has an initial feature vector $\mathbf{x}_i^{(0)} \in \mathbb{R}^{d_n}$, and each $p$-th edge from $i$ to $j$ has a feature vector $\mathbf{e}_{ijp}^{(0)} \in \mathbb{R}^{d_e}$ and $p \in [P_{ij}]$. At the $l$-th layer where $l \in [L]$ and $L$ is the total number of layers, the latent node and edge features are denoted $\mathbf{x}_i^{(l)} \in \mathbb{R}^d$ and $\mathbf{e}_{ijp}^{(l)} \in \mathbb{R}^d$.

Let $\mathcal{M}_d$ denote the space of multisets over $\mathbb{R}^d$, and let $\mathcal{M}(\mathcal{M}_d)$ denote the space of multisets of such multisets. Let $j \in \mathcal{V}$ be a target node and, $X_{ij} = \{\{\mathbf{e}_{ijp} \mid p \in [P_{ij}]\}\} \in \mathcal{M}_d$, denote the multiset of edge feature vectors from node $i$ to node $j$. The neighborhood of $j$ is then given as $\mathcal{X}_j = \{\{X_{ij} \mid i \in N_{\mathrm{in}}(j)\}\} \in \mathcal{M}(\mathcal{M}_d)$.

Let $g_1, \ldots, g_k : \mathcal{M}_d \to \mathbb{R}^d$. be a collection of coordinate-wise aggregators. Analogous to PNA (Corso et al., 2020), we define an aggregation function, $f_\theta$, that applies each aggregator to the input, concatenates the results, and processes the concatenated vector through an MLP:

$$f_\theta : \mathcal{M}_d \to \mathbb{R}^{d'}, \quad f_\theta(X) := \mathrm{MLP}_\theta([g_1(X) \, \| \, \ldots \, \| \, g_k(X)]), \tag{1}$$

where $X \in \mathcal{M}_d$, $\|$ is concatenation, $\mathrm{MLP}_\theta : \mathbb{R}^{kd} \to \mathbb{R}^{d'}$ is a feedforward network and $d'$ denotes the output dimension of the MLP.

## 3.2 SINGLE-STAGE VS TWO-STAGE AGGREGATION

In many real-world graphs, such as financial transaction networks, multiple edges may connect the same pair of nodes, each with distinct attributes. Standard GNNs, however, typically ignore this edge multiplicity and apply *single-stage aggregation*, which aggregates all incoming edges at once.

**Definition 3** (Single-stage Aggregation). *A single-stage aggregation function $\mathcal{T}_{\mathrm{single\text{-}stage}}$ : $\mathcal{M}(\mathcal{M}_d) \to \mathbb{R}^d$ aggregates all edge features in the neighborhood $\mathcal{X}_j$, treating it as a single multiset.*

$$\mathcal{T}_{\mathrm{single\text{-}stage}}(\mathcal{X}_j) := f_\theta \Big( \biguplus_{X_{ij} \in \mathcal{X}_j} X_{ij} \Big). \tag{2}$$

In standard GNNs, $f_\theta$ is commonly implemented using a single aggregation function $g$, i.e., $k = 1$ in Equation 1, where $g$ is typically chosen as SUM, MEAN, or MAX.

Crucially, in multigraphs, single-stage aggregation fails to distinguish between edges from the same neighbor and those from different neighbors. To address this, we propose a *two-stage aggregation* scheme: first, features of multi-edges between the same node pair are aggregated; second, the resulting messages from distinct neighbors are aggregated at the node level.

**Definition 4** (Two-stage Aggregation). *A two-stage aggregation function $\mathcal{T}_{\mathrm{two\text{-}stage}} : \mathcal{M}(\mathcal{M}_d) \to \mathbb{R}^d$ first aggregates each $X_{ij} \in \mathcal{X}_j$ individually, and then aggregates the resulting multiset of vectors.*

$$\mathcal{T}_{\mathrm{two\text{-}stage}}(\mathcal{X}_j) := f_{\theta_2} \left( \{\{f_{\theta_1}(X_{ij}) \mid X_{ij} \in \mathcal{X}_j\}\} \right), \tag{3}$$

This two-stage aggregation scheme naturally distinguishes edges based on their source nodes, enabling the computation of per-neighbor statistics. By applying multiple aggregators at both stages, as defined in Equation 1, we can extract more nuanced statistical information, capturing both per-neighbor and overall neighborhood characteristics. We formalize the enhanced representational capacity of this approach under a class of moment-based aggregators, inspired by (Corso et al., 2020). Specifically, we employ the sum and raw moments to capture rich distributional features of multisets.

**Theorem 1.** *Two-stage aggregation induces a strictly larger image than single-stage aggregation, if both schemes use the same set of $k$ aggregators: the sum and the raw moments of orders $2$ through $k$, defined as*

$$g_1(X) := \sum_{x \in X} x, \quad g_r(X) := \frac{1}{|X|} \sum_{x \in X} x^r, \quad 2 \le r \le k,$$

*where $X \in \mathcal{M}_d$ and $x^r$ is element-wise $r$-th power.*

The proof of Theorem 1 is in Appendix B.1. We first show that two-stage aggregation can replicate single-stage aggregation by computing neighborhood moments from per-neighbor moments. Furthermore, we show that single-stage aggregation fails to capture per-neighbor moments, whereas two-stage aggregation inherently does, allowing the extraction of more nuanced statistical information from the neighborhood.

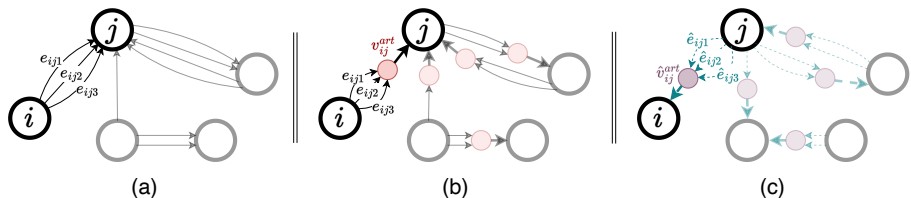

Figure 2: **Illustration of Two-stage Aggregation with artificial nodes**. (a) A multigraph with multi-edges $e_{ij1}, e_{ij2}, e_{ij3}$ between nodes $i$ and $j$. (b) Illustration of *artificial nodes*, which helps us to compute latent features of aggregated multi-edges between adjacent node pairs. First, features of multi-edges are aggregated at the artificial nodes; next, each destination node aggregates messages from its neighboring artificial nodes. (c) The bi-directional message passing mechanism. In directed multigraphs, reverse edges are added in the opposite direction of the original edges. Separate message computations are performed to handle the original and reversed edges.

## 3.3 THE MEGA-GNN ARCHITECTURE

This section introduces a novel message passing architecture for multigraphs with edge attributes that implements the two-stage aggregation scheme outlined in Section 3.2. For notational convenience, we introduce *artificial nodes*, which help us compute latent features of aggregated multi-edges between adjacent node pairs (see Figure 2(b)): $\mathcal{V}^{art} = \{v_{ij}^{art} \mid (i,j) \in \mathcal{E}^{supp}\}$.

Artificial nodes serve as temporary computational intermediates during aggregation and do not alter the graph topology as shown in Figure 3. Let $\mathbf{h}_{ij}^{(l-1)} \in \mathbb{R}^d$ be the $d$-dimensional latent feature vector of the artificial node $v_{ij}^{art}$. Recall that $X_{ij} \in \mathcal{M}_d$ denotes the multiset of edge features between node $i$ and node $j$. At layer $(l-1)$, the corresponding multiset of latent edge features are $X_{ij}^{(l-1)} = \{\{\mathbf{e}_{ijp}^{(l-1)} \mid p \in [P_{ij}]\}\}$.

In the first aggregation stage, latent features of multi-edges are aggregated at the artificial nodes.

$$\mathbf{h}_{ij}^{(l-1)} = f_{\theta_1}^{(l-1)}\big(X_{ij}^{(l-1)}\big), \tag{4}$$

where $f_{\theta_1}^{(l-1)} : \mathcal{M}_d \to \mathbb{R}^d$. In the second stage, node-level aggregation is performed over messages from artificial nodes :

$$\mathbf{a}_j^{(l-1)} = f_{\theta_2}^{(l-1)}\big(\{\{[\mathbf{x}_i^{(l-1)} \| \mathbf{h}_{ij}^{(l-1)}] \mid i \in N_{in}(j)\}\}\big), \tag{5}$$

$$\mathbf{x}_j^{(l)} = \phi_n^{(l-1)}\big([\mathbf{x}_j^{(l-1)} \| \mathbf{a}_j^{(l-1)}]\big), \tag{6}$$

where $\|$ is concatenation, $f_{\theta_2}^{(l-1)} : \mathcal{M}_{2d} \to \mathbb{R}^d$ and $\phi_n^{(l-1)} : \mathbb{R}^{2d} \to \mathbb{R}^d$ is the node update function at layer $(l-1)$. Here, the destination node $j$ receives a single (aggregated) message from each of its distinct incoming neighbors. Then, the latent features of each edge are updated:

$$\mathbf{e}_{ijp}^{(l)} = \phi_e^{(l-1)}\big([\mathbf{x}_i^{(l-1)} \| \mathbf{e}_{ijp}^{(l-1)} \| \mathbf{h}_{ij}^{(l-1)}]\big), \tag{7}$$

where $\|$ is concatenation and $\phi_e^{(l-1)} : \mathbb{R}^{3d} \to \mathbb{R}^d$ is the edge update function.

Notably, MEGA-GNN preserves the original multigraph topology while enabling joint propagation of node and edge latent features at each layer. Unlike ADAMM (Sotiropoulos et al., 2023), it maintains distinct edges through individual updates. A detailed architecture diagram of the proposed method is provided in Figure 3.

## 3.4 MEGA-GNN WITH BI-DIRECTIONAL MESSAGE PASSING

Bi-directional message passing improves model capacity by aggregating messages from incoming and outgoing neighbors separately. For example, it enables the computation of both the in-degree and the out-degree of a node, which is not possible using only incoming messages or by treating the graph as undirected (Egressy et al., 2024). This subsection describes the way MEGA-GNN implements bi-directional message passing in combination with two-stage aggregations.

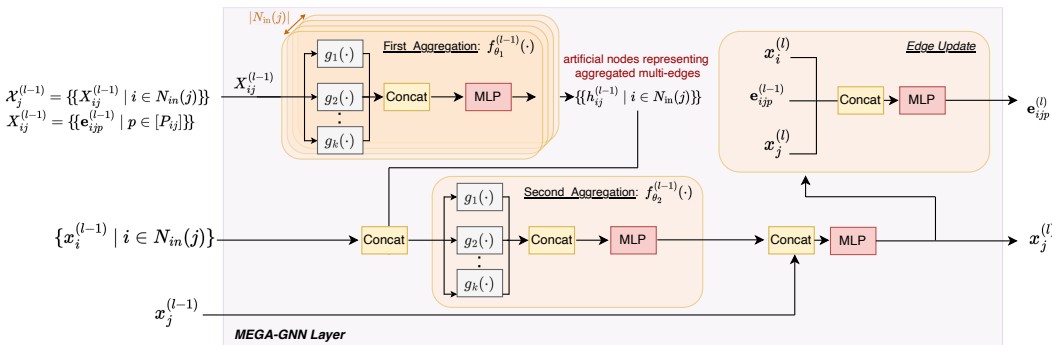

Figure 3: Overview of the MEGA-GNN layer. $g_1(\cdot), \dots, g_k(\cdot)$ are aggregation functions.

Formally, we define reversed edges $(j, i)$ for each original edge $(i, j) \in \mathcal{E}$, and initialize their features as $\hat{\mathbf{e}}_{ijp}^{(0)} := \mathbf{e}_{jip}^{(0)}$, where $p \in [P_{ji}]$. Similarly to Section 3.3, we use the notion of *artificial nodes* to compute latent features of aggregated reversed multi-edges between adjacent node pairs (Figure 2(c)): $\hat{\mathcal{V}}^{art} = \{\hat{v}_{ij}^{art} \mid (j, i) \in \mathcal{E}^{supp}\}$.

Let $\hat{\mathbf{h}}_{ij}^{(l-1)} \in \mathbb{R}^d$ be the $d$-dimensional latent feature vector of the artificial node $\hat{v}_{ij}^{art}$. At layer $(l-1)$, we denote the multiset of latent edge features from node $j$ to $i$ as $\hat{X}_{ij}^{(l-1)} = \{\{\hat{\mathbf{e}}_{ijp}^{(l-1)} \mid p \in [P_{ji}]\}\}$.

In the first stage, the latent features of outgoing multi-edges from node $j$ to $i$ are aggregated.

$$\hat{\mathbf{h}}_{ij}^{(l-1)} = \hat{f}_{\theta_1}^{(l-1)}(\hat{X}_{ij}^{(l-1)}), \tag{8}$$

where $\hat{f}_{\theta_1}^{(l-1)} : \mathcal{M}_d \to \mathbb{R}^d$. In the second stage, node-level aggregation is performed.

$$\hat{\mathbf{a}}_j^{(l-1)} = \hat{f}_{\theta_2}^{(l-1)}\big(\{\{[\mathbf{x}_i^{(l-1)} \parallel \hat{\mathbf{h}}_{ij}^{(l-1)}] \mid i \in N_{out}(j)\}\}\big) \tag{9}$$

$$\mathbf{x}_j^{(l)} = \hat{\phi}_n^{(l-1)}\bigg([\mathbf{x}_j^{(l-1)} \parallel \mathbf{a}_j^{(l-1)} \parallel \hat{\mathbf{a}}_j^{(l-1)}]\bigg), \tag{10}$$

where $\parallel$ is concatenation, $\hat{f}_{\theta_2}^{(l-1)} : \mathcal{M}_{2d} \to \mathbb{R}^d$, $\hat{\phi}_n^{(l-1)} : \mathbb{R}^{3d} \to \mathbb{R}^d$ is the node update function at layer $(l-1)$ and $\mathbf{a}_j^{(l-1)}$ is computed using Equation 5. Thus, messages from incoming and outgoing neighbors are aggregated separately, and combined to update the destination node $j$. Similarly, the latent features of the reverse edges are updated with function $\hat{\phi}_e^{(l-1)} : \mathbb{R}^{3d} \to \mathbb{R}^d$:

$$\hat{\mathbf{e}}_{ijp}^{(l)} = \hat{\phi}_e^{(l-1)}\big([\mathbf{x}_i^{(l-1)} \parallel \hat{\mathbf{e}}_{ijp}^{(l-1)} \parallel \hat{\mathbf{h}}_{ij}^{(l-1)}]\big), \tag{11}$$

A detailed architecture diagram of the proposed method with bi-directional message passing capability is provided in Appendix C.1.

## 3.5 Additional Properties of MEGA-GNN

### 3.5.1 Permutation Equivariance

**Definition 5** (Permutation Equivariance). *A function $\psi$ is permutation equivariant with respect to node and edge permutations if, for any permutation $\rho$ acting on the nodes and edges of a graph $\mathcal{G} = (\mathcal{V}, \mathcal{E})$, the following holds: $\psi(\rho \circ \mathcal{G}(\mathcal{V}, \mathcal{E})) = \rho \circ \psi(\mathcal{G}(\mathcal{V}, \mathcal{E}))$.*

**Proposition 1** (Permutation Equivariance). *Given aggregation functions $f_{\theta_1}$ and $f_{\theta_2}$ that are permutation invariant over multisets, MEGA-GNN is permutation equivariant with respect to arbitrary permutations of nodes and edges in the input multigraph, including permutations over multi-edges.*

Proposition 1 (proof in Appendix B.3) shows MEGA-GNN maintains permutation equivariance at both node and edge levels, a property not shared by Multi-GNN (Egressy et al., 2024), as stated in Proposition 2 of Appendix A.2.

### 3.5.2 UNIVERSALITY

**Definition 6** (Universality). *An MPNN is universal if it can approximate every invariant or equivariant continuous function defined on graphs (Keriven & Peyré, 2019; Loukas, 2020).*

**Theorem 2.** *Let $\mathcal{G} = (\mathcal{V}, \mathcal{E})$ be a connected directed multigraph with $m = |\mathcal{E}|$. Given an injective edge-labeling function $L : \mathcal{E} \to [M]$ for some $M \geq m$ that assigns a unique label to each edge, MEGA-GNN can compute unique node IDs on $\mathcal{G}$.*

The proof of Theorem 2 is given in Appendix B.4. The proof relies on the existence of an injective edge-labeling function $L : \mathcal{E} \to [M]$, derived from the edge features. We demonstrate that MEGA-GNN can leverage these distinct edge labels to generate unique node identifiers.

**Corollary 1** (Universality). *Let $\mathcal{G} = (\mathcal{V}, \mathcal{E})$ be a connected directed multigraph with $m = |\mathcal{E}|$. Assume that there exists an injective edge-labeling function, $L : \mathcal{E} \to [M]$ for some $M \geq m$, that assigns a unique label to each edge. Then, given enough layers with sufficient expressiveness and width, MEGA-GNN is universal.*

*Proof.* As outlined by Loukas (2020), the following conditions must be met to achieve universality: a sufficient number of layers, layers with adequate expressive power and width, and the ability for nodes to uniquely distinguish one another. Theorem 2 shows that MEGA-GNN can uniquely distinguish nodes in connected multigraphs. In addition, MEGA-GNN can be configured to incorporate a sufficient number of linear and nonlinear layers with adequate expressive power and width. Therefore, MEGA-GNN satisfies all the necessary conditions and is universal based on Definition 6. □

Crucially, MEGA-GNN's ability to assign unique node IDs does not conflict with its permutation equivariance property (see Proposition 1). In Theorem 2, we assume the existence of injective edge-labeling function, which serves as foundation for our theoretical analysis in Corollary 1. In practice, MEGA-GNN treats edge features as static inputs, without enforcing any canonical ordering.

### 3.5.3 INFERENCE COMPLEXITY

**Theorem 3.** *The asymptotic complexity of a single MEGA-GNN layer is:*

$$O\big((|\mathcal{E}| + |\mathcal{V}|)d^2 + (|\mathcal{E}| + |\mathcal{V}|)d\big).$$

In Theorem 3 (proof is in Appendix B.5), the first term arises from linear layers used in feature transformations, while the second term accounts for neighborhood aggregation and element-wise nonlinearities. This shows that the two-stage aggregation does not increase the asymptotic complexity: MEGA-GNN matches the complexity of standard message-passing GNNs with edge updates.

## 4 EXPERIMENTS

This section presents experiments to evaluate the accuracy of MEGA-GNN across multiple tasks. We focus on two key applications in financial crime analysis: money laundering detection via edge classification and phishing account detection via node classification. Additionally, we evaluate MEGA-GNN on three temporal user-item interaction datasets for the user state-change prediction task. In total, we assess the performance of MEGA-GNN on ten distinct datasets spanning three different tasks, as described below. Further dataset statistics and details are provided in Appendix C.4.

**Anti-Money Laundering (AML)**: We use IBM's realistic synthetic financial transaction datasets for the money laundering detection task (Altman et al., 2023). There are six AML datasets in total: small, medium, and large variants, each available in two versions, LI (low illicit ratio) and HI (high illicit ratio). The small, medium, and large datasets contain approximately 6 million, 30 million, and 180 million transactions, respectively. The LI versions have an illicit transaction rate of around $0.05\%$, while the HI versions have a slightly higher rate of approximately $0.1\%$. The task is framed as edge classification, where each transaction must be labeled as either illicit or non illicit. Edges in the transaction graph include four attributes: timestamp, amount, currency, and payment format.

**Ethereum Phishing Transaction Network (ETH)**: Since access to real financial transaction data from banks is limited, cryptocurrencies provide an alternative data source. In our study, we use a

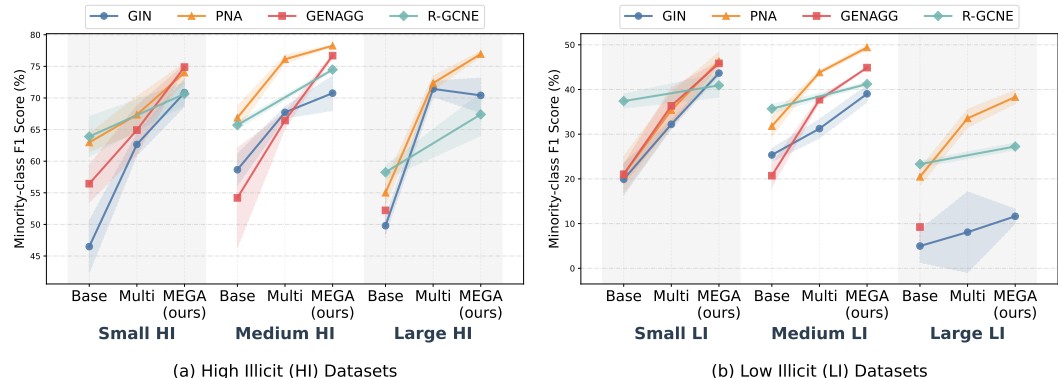

Figure 4: Minority class F1 scores (%) for six AML datasets using four different GNN baselines (GIN, PNA, GenAgg, and R-GCNE) and two different multigraph adaptations (Multi and MEGA).

real-world transaction graph extracted from the Ethereum blockchain (Chen et al., 2021a), where accounts are treated as nodes and transactions as edges. Each node has a label that indicates whether it is a phishing node. The edges have two attributes: timestamp and amount.

**User State-Change Prediction**: We use three temporal user-item interaction datasets from JODIE (Kumar et al., 2019) for this task: Reddit bans, Wikipedia bans, and MOOC dropouts, all derived from social and collaborative platforms. Each interaction is labeled 0 until a user is banned or drops out; their final interaction is labeled 1.

**Implementation**: We implement our solutions using PyTorch Geometric (Fey & Lenssen, 2019). For the AML datasets, we use the same temporal splits as Multi-GNN. For ETH, we use a 65/15/20 temporal split. For the JODIE datasets, we follow the experimental setup in (Kumar et al., 2019) and use a 60/20/20 temporal split. We adopt ego IDs (You et al., 2021) only in edge classification experiments (Figure 4, Table 4) and incorporate bi-directional message passing in both edge and node classification experiments (Figure 4, Tables 4 and 1). To ensure statistical significance, each experiment is repeated at least five times with different random seeds and the mean and standard deviation of these runs are reported. Additional implementation details are given in Appendix C.

**Evaluation:** We evaluate four baseline architectures (GIN, PNA, GenAgg, R-GCN) in combination with three multigraph adaptations (Multi-GNN Egressy et al. (2024), ADAMM Sotiropoulos et al. (2023), and MEGA-GNN). The *Base* models do not use multigraph adaptations, but leverage edge updates by default. GIN provides a basic GNN architecture with expressiveness guarantees. PNA exhibits strong empirical performance on multigraphs (Egressy et al., 2024). GenAgg uses a learnable aggregator that can parameterize common aggregators (Kortvelesy et al., 2023). In addition to these GNN baselines, we evaluate Multi-FraudGT (Lin et al., 2024), a graph-transformer that incorporates multigraph adaptations of Multi-GNN (Egressy et al., 2024) and DIAM (Ding et al., 2024) a specialized solution for the ETH benchmark.

In our experiments we first extend R-GCN to incorporate edge attributes and edge updates, referring to this variant as R-GCNE (see Appendix C.6). To apply R-GCNE to the AML datasets, we assign edge types based on transaction currency, and resulting edges carry both relation types and attributes. However, such a conversion is not always possible; for instance, the ETH dataset lacks categorical features suitable for defining relation types. When both forms of information are present (relation types and attributes), MEGA-GNN can be combined with R-GCNE to construct MEGA-R-GCNE (see Appendix C.5), although MEGA can be paired with any GNN baseline.

**AML Edge Classification:** Figure 4 presents minority-class F1 scores across six synthetic AML datasets, comparing GNN baselines with their multigraph adaptations (see Appendix E for more detailed results). Note that we extended R-GCNE to support only MEGA adaptations. In addition, Multi-GenAgg and MEGA-GenAgg experiments exceeded GPU memory size for Large datasets.

In all 22 cases, MEGA-GNNs significantly outperform baseline GNNs. In 15 of the 16 comparison points, MEGA-GNNs also outperform Multi-GNNs. On average, MEGA-GNNs improve minority-class F1 scores by 4.75% on HI datasets and by 6.77% on LI datasets compared to Multi-GNNs, es-

Table 1: ETH node classification: minority-class F1 scores (%). The best result is given in **boldface**.

| | ADAMM | | Multi | | | DIAM | MEGA (ours) | |
| --- | --- | --- | --- | --- | --- | --- | --- | --- |
| | GIN | PNA | GIN | PNA | FraudGT | | GIN | PNA |
| F1 | $34.73 \pm 15.75$ | $37.99 \pm 5.41$ | $51.34 \pm 3.92$ | $\underline{64.61 \pm 1.40}$ | $57.40 \pm 0.91$ | $64.43 \pm 1.07$ | $57.45 \pm 1.14$ | $\mathbf{64.84 \pm 1.73}$ |

tablishing new state-of-the-art results. These results demonstrate the effectiveness of MEGA-GNN in detecting complex financial crime patterns in large-scale transaction networks. Moreover, our consistent gains across different baseline architectures indicate that MEGA-GNN's improvements are not architecture dependent, but stem from its novel two-stage message aggregation mechanism, which enables computation of more nuanced statistics on multigraphs, as shown in Theorem 1.

Notably, we also observe that MEGA adaptations consistently improve the R-GCNE results as well. This shows that two techniques are are orthogonal yet complementary. While R-GCNE is designed for labeled multigraphs with predefined relation types, MEGA-GNN targets edge-attributed multigraphs with high-dimensional features. When edge relation types and attributes are both available, MEGA-R-GCNE effectively leverages them, combining the strengths of both techniques.

**ETH Node Classification:** Table 1 presents our experiments on the ETH dataset (Chen et al., 2021a). MEGA-GIN improves the minority class F1 score by 6.11% over Multi-GIN and MEGA-PNA delivers the highest F1 score, achieving a slight improvement over Multi-PNA (Egressy et al., 2024) and DIAM (Ding et al., 2024). When compared to ADAMM (Sotiropoulos et al., 2023), our MEGA-GNN variants exhibit a striking improvement, consistently delivering over 20% higher performance. Furthermore, compared to Multi-FraudGT, MEGA-PNA achieves a 7.44% higher performance. These results confirm the effectiveness of MEGA-GNN's two-stage message aggregation mechanism and its bi-directional message passing capabilities.

Table 2: Impact of permuting port numbers on the F1 scores (%) of Multi-FraudGT and Multi-PNA.

| Ablation | AML Small HI | AML Small LI | AML Medium HI | AML Medium LI | AML Large HI | AML Large LI | ETH |
| --- | --- | --- | --- | --- | --- | --- | --- |
| Multi-PNA | $67.35 \pm 2.89$ | $35.40 \pm 3.93$ | $\underline{76.13 \pm 0.69}$ | $43.82 \pm 0.51$ | $72.35 \pm 1.14$ | $33.54 \pm 2.04$ | $64.61 \pm 1.40$ |
| Multi-PNA (permuted) | $63.77 \pm 2.47$ | $31.48 \pm 0.72$ | $73.36 \pm 0.83$ | $43.24 \pm 0.24$ | $70.93 \pm 0.69$ | $32.18 \pm 1.72$ | $62.71 \pm 2.73$ |
| Multi-FraudGT | $\mathbf{75.81 \pm 0.75}$ | $\underline{45.69 \pm 1.14}$ | $75.97 \pm 0.18$ | $\underline{44.66 \pm 0.58}$ | $\underline{73.04 \pm 0.59}$ | $\underline{35.49 \pm 0.52}$ | $57.40 \pm 0.91$ |
| Multi-FraudGT (permuted) | $61.74 \pm 1.68$ | $30.15 \pm 2.67$ | $65.89 \pm 5.61$ | $32.05 \pm 1.35$ | $63.33 \pm 1.35$ | $29.95 \pm 1.18$ | $49.59 \pm 1.83$ |
| MEGA-PNA | $\underline{74.01 \pm 1.55}$ | $\mathbf{46.32 \pm 2.07}$ | $78.26 \pm 0.11$ | $\mathbf{49.40 \pm 0.54}$ | $\mathbf{76.95 \pm 0.44}$ | $\mathbf{38.31 \pm 1.53}$ | $\mathbf{64.84 \pm 1.73}$ |

**Permutation Equivariance:** Architectures using Multi-GNN adaptations, such as Multi-PNA and Multi-FraudGT, are not permutation equivariant, as they rely on precomputed port numbers. Table 2, shows that MEGA-PNA outperforms both architectures in most scenarios. Notably, Table 2 shows that permuting port numbers during testing decreases the F1 scores of Multi-PNA and Multi-FraudGT considerably and further increases the gap between MEGA-PNA and these two models.

**Ablations:** Table 3 evaluates the standalone impact of two-stage aggregation mechanism of our proposed approach (Sec. 3.3) on top of different baseline architectures (GIN,PNA), then analyzes the effects of adding bi-directional message passing (Sec. 3.4) and ego IDs (You et al., 2021). The results indicate that the primary performance improvements of MEGA-GNN are driven by the two-stage aggregation itself. Bi-directional MP provides further gains, particularly on the ETH dataset, while ego IDs offer selective improvements.

Table 3: Impact of bi-directional message passing (MP) and ego IDs: minority-class F1 scores (%) of MEGA-GNN.

| Ablation | AML Small HI | AML Small LI | ETH |
| --- | --- | --- | --- |
| GIN | $46.50 \pm 4.11$ | $19.93 \pm 3.55$ | $42.33 \pm 3.70$ |
| MEGA-GIN (GIN with Two-stage Agg.) | $69.98 \pm 2.02$ | $41.45 \pm 2.13$ | $43.56 \pm 2.67$ |
| MEGA-GIN w/ Bi-directional MP | $\mathbf{72.50 \pm 3.26}$ | $\underline{41.67 \pm 1.51}$ | $\mathbf{57.45 \pm 1.14}$ |
| MEGA-GIN w/ ego IDs & Bi-directional MP | $\underline{70.83 \pm 2.18}$ | $\mathbf{43.66 \pm 0.54}$ | $55.19 \pm 2.33$ |
| PNA | $62.96 \pm 1.43$ | $21.02 \pm 4.05$ | $53.93 \pm 2.45$ |
| MEGA-PNA (PNA with Two-stage Agg.) | $73.65 \pm 0.36$ | $43.77 \pm 1.53$ | $59.13 \pm 0.51$ |
| MEGA-PNA w/ Bi-directional MP | $\mathbf{74.98 \pm 1.59}$ | $\underline{45.36 \pm 1.18}$ | $\mathbf{64.84 \pm 1.73}$ |
| MEGA-PNA w/ ego IDs & Bi-directional MP | $\underline{74.01 \pm 1.55}$ | $\mathbf{46.32 \pm 2.07}$ | $60.02 \pm 5.10$ |

**User State-Change Prediction:** Table 4 compares MEGA-PNA with JODIE, PNA, and Multi-PNA on three temporal user-item interaction datasets. On MOOC, MEGA-PNA slightly outper-

forms JODIE; on Wikipedia, it achieves the highest ROC-AUC, surpassing all baselines; and on Reddit, a highly imbalanced benchmark with only 366 positives among 672,447 interactions, MEGA-GNN delivers a clear gain. These results demonstrate the applicability of MEGA-GNN beyond the analysis of financial transaction networks.

Table 4: User state-change prediction performance (ROC-AUC, %) on three temporal user-item interaction datasets Kumar et al. (2019) derived from social and collaborative platforms.

| Method | MOOC | Wikipedia | Reddit |
|---|---|---|---|
| JODIE | 75.6 ± na | 83.1 ± na | 59.9 ± na |
| PNA | 66.3 ± 3.3 | 76.7 ± 2.5 | 63.5 ± 5.9 |
| Multi-PNA | 70.1 ± 3.6 | 90.4 ± 1.3 | 61.6 ± 5.7 |
| MEGA-PNA | **76.1** ± 1.8 | **92.1** ± 0.9 | **67.6** ± 2.8 |

Table 5: Inference runtime analysis on AML Small HI dataset.

| Model | F1 | Original Model Sizes | Matched Parameters (∼160k) |
|---|---|---|---|
| | | Inference Time / Epoch (s) | Inference Time / Epoch (s) |
| GIN | 46.50 ± 4.11 | 1.21 | 2.01 |
| MEGA-GIN (GIN with Two-stage Agg.) | 69.98 ± 2.02 | 1.51 | 2.23 |
| MEGA-GIN w/ Bi-directional MP | 72.50 ± 3.26 | 6.77 | 6.77 |
| MEGA-GIN w/ ego IDs & Bi-directional MP | 70.83 ± 2.18 | 6.82 | 6.82 |

**Inference Runtime Analysis:** Table 5 presents the inference runtimes of the models investigated in our ablation study in Table 3. While the addition of two-stage aggregation to the GIN baseline increases the inference time per epoch by 24.7%, it yields a substantial 50.4% gain in F1-score. Furthermore, this overhead drops to 10.9% when we match the parameter counts. As expected, bi-directional message passing incurs a higher cost because it requires propagation over both original and reversed edges. Conversely, the inclusion of ego IDs adds negligible overhead. In addition, Table 11 of Appendix D compares the inference throughput rate of MEGA-GNN and Multi-GNN variants, showing that the runtime overhead of MEGA-GNN's two-stage aggregation is minimal compared to Multi-GNN's single-stage aggregation.

## 5 CONCLUSION

We introduce MEGA-GNN, a GNN architecture that leverages a novel two-stage message aggregation mechanism for edge-attributed multigraphs. MEGA-GNN is built on a rigorous theoretical foundation: its two-stage aggregation is strictly more expressive than standard single-stage aggregation schemes. Furthermore, MEGA-GNN ensures permutation equivariance: unlike the prior methods Egressy et al. (2024); Lin et al. (2024), MEGA-GNN does not rely on pre-computed multigraph port identifiers. Additionally, MEGA-GNN achieves universality when there exists a bijective edge-labeling (e.g., via timestamps). We show that MEGA-GNN integrates seamlessly with diverse GNN baselines such as GIN, PNA, GenAgg, and R-GCN. Importantly, MEGA-GNN's asymptotic inference complexity matches that of the baseline GNNs that incorporate edge updates.

The effectiveness of MEGA-GNN is validated by empirical evaluations on data sets combining multigraph structure, complex topological patterns, edge attributes, and extreme class imbalance. On financial crime datasets, MEGA-GNNs achieve substantial improvements on average: 16.57% higher minority-class F1 scores than baseline GNNs, 4.98% higher than Multi-GNNs, and 2.86% higher than Multi-FraudGT. When port numbers are permuted at test time, the performance gap widens: MEGA-GNNs outperform Multi-GNNs by 7.2% and Multi-FraudGT by 13.62% on average in minority-class F1 score. Lastly, on temporal user-item interaction datasets, MEGA-PNA outperforms JODIE Kumar et al. (2019) by 5.73% and Multi-PNA by 4.56% on average in ROC-AUC score, demonstrating MEGA-GNN's broader applicability.

**Limitations & Future Work:** MEGA-GNN has so far been applied only to financial transaction datasets and temporal user–item interaction datasets from social platforms. In future work, we plan to extend our approach to property graphs with edge properties, transportation networks, and cybersecurity datasets.

REPRODUCIBILITY STATEMENT

We include the source code in the supplementary materials to enable reproduction of all experimental results. The codebase provides clear instructions for downloading and preprocessing datasets, setting up the environment, and running experiments. For each model variant considered in the experiments, we supply a separate configuration file that specifies the corresponding hyperparameter settings. Additional details on model architectures, training procedures, and evaluation protocols are given in Section 4 and Appendix C.

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

## A    LIMITATIONS OF EXISTING SOLUTIONS

### A.1    MULTI-RELATIONAL GNNS

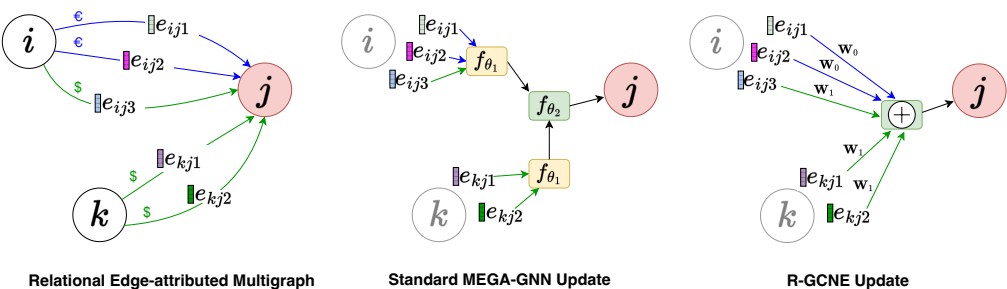

**Relational Edge-attributed Multigraph**   **Standard MEGA-GNN Update**   **R-GCNE Update**

Figure 5: Comparison of MEGA-GNN and R-GCNE architectures. MEGA-GNN performs a two-stage aggregation: it first aggregates multiple edges between the same node pair (edge-level aggregation) and then applies node-level aggregation. In contrast, R-GCNE performs single stage aggregation, and it applies relation-specific transformations by multiplying each edge feature with a learnable weight matrix based on its relation type (currency in the example graph). For clarity, inverse relations and self-loops are omitted in the R-GCNE illustration.

Relational Graph Convolutional Networks (R-GCNs) Schlichtkrull et al. (2018) are specifically designed for *labeled* multigraphs, where each edge is assigned a relation type from a fixed, finite set of discrete labels. R-GCNE (see Appendix C.6) is an extension of R-GCN Schlichtkrull et al. (2018) that uses edge attributes and edge updates. R-GCNE achieves relation-aware message passing by applying distinct learnable weight matrices for each relation type. However, despite this relational specificity, R-GCNE performs a *single-stage aggregation*: messages from all neighbors are summed, optionally scaled by a problem-specific normalization constant. With single-aggregation scheme R-GCNE cannot distinguish between edges originating from the same neighbor (i.e. multi-edges) and edges originating from different neighbors. By contrast, MEGA-GNN introduces a novel two-stage aggregation mechanism. First, it aggregates the attributes of multiple edges connecting the same pair of nodes (edge-level aggregation), capturing intra-pair interactions and edge-specific statistics. Second, the resulting per-neighbor representations are aggregated across distinct neighbors (node-level aggregation). As shown in Theorem 1, this hierarchical design allows MEGA-GNN to compute detailed per-neighbor statistics that standard message-passing GNNs, including R-GCNE, inherently overlook. Such capability is critical in financial transaction networks. A visual comparison of these two architectures is provided in Figure 5, where inverse relations and self-loops are omitted from the R-GCNE diagram for simplicity.

To apply R-GCN to Anti-Money Laundering (AML) datasets in our experiments in Section 4, we converted the multigraph into a multi-relational graph by assigning edge types based on transaction currency. However, this transformation is not always feasible; for example, in the ETH dataset (see Section 4), multigraphs often lack well-defined relation types, limiting the applicability of standard relational GNNs. Importantly, our work is orthogonal to existing relational GNN approaches and naturally extends to multi-relational multigraphs—that is, graphs where multiple edges of the same type exist between the same node pair. This extension enables the development of a hybrid model, MEGA-R-GCNE, which integrates our multi-edge aggregation strategy with R-GCNE-like architectures, combining the strengths of both approaches. An illustration of the hybrid method is shown in Figure 6, with experimental results presented in Section 4.

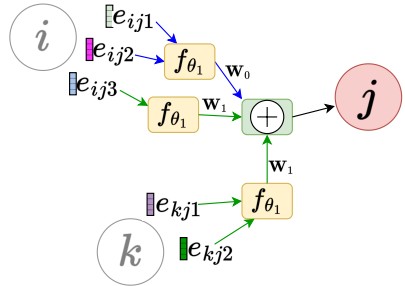

**MEGA-R-GCN Update**

Figure 6: Illustration of MEGA-R-GCNE.

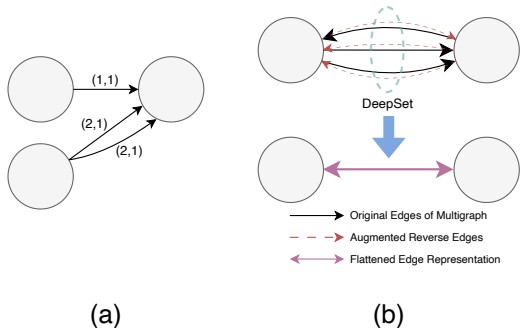

(a)                  (b)

Figure 7: (a) Directed multigraph port numbering of Multi-GNN Egressy et al. (2024). (b) Illustration of multigraph to simple graph transformation by ADAMM Sotiropoulos et al. (2023)

## A.2 MULTI-GNN, ADAMM AND DIAM

Table 6: Related work vs. MEGA-GNN. MP refers to Message Passing, Aggr. stands for Aggregation.

| Features | Multi-GNN | ADAMM | DIAM | MEGA-GNN |
|---|---|---|---|---|
| Proof of Universality | ✓ | | | ✓ |
| Bi-directional MP | ✓ | | ✓ | ✓ |
| Edge Embeddings | ✓ | | | ✓ |
| Node Embeddings | ✓ | ✓ | ✓ | ✓ |
| Permutation Equivariance | | ✓ | ✓ | ✓ |
| Two-stage Aggr. | | ✓ | | ✓ |
| Two-stage Aggr. in MP | | | | ✓ |
| Proof of Two-stage is more powerful | | | | ✓ |

In the literature, two key works specifically address multigraphs: Multi-GNN Egressy et al. (2024) and ADAMM Sotiropoulos et al. (2023).

**Multi-GNN** introduced a provably powerful GNN architecture for directed multigraphs, incorporating simple adaptations such as reverse message passing, port numbering, and ego IDs You et al. (2021). A notable contribution of Multi-GNN is the multigraph port numbering, which enables the model to distinguish between edges originating from the same neighbor and those from different neighbors (see Figure A.2 (a)). These three adaptations make it possible to assign unique node IDs in connected directed multigraphs, making the Multi-GNN solution universal. However, augmenting edge features with port numbers results in the loss of permutation equivariance (see Proposition 2 and the proof in Appendix B.2). This loss is significant because permutation equivariance is a crucial property for ensuring that the model's predictions remain consistent under arbitrary permutations of nodes or edges in graph learning tasks. Empirical evaluation on the impact of permuting port numbers during inference is presented in Table 2.

**Definition 7** (Strict Total Order). *Strict total order is a binary relation $<$ on a set $\mathcal{A}$, which satisfies the following conditions for all $a,b,c \in \mathcal{A}$: (1) if $a \neq b$, either $a < b$ or $b < a$, (2) not $a < a$, (3) if $a < b$ and $b < c$, then $a < c$ (see (Munkres, 2000), p. 22).*

**Proposition 2.** *The multigraph port numbering Egressy et al. (2024), is not permutation equivariant in the absence of a strict total ordering of edges.*

**ADAMM** aggregates multi-edges between two nodes into a single undirected super-edge (see Figure A.2 (b)), before message passing layers. The initial features for this super-edge are computed using DeepSet Zaheer et al. (2017), incorporating the direction of the edge as an additional edge feature to differentiate between original and augmented reverse edges. The subsequent message passing layers then operate on these aggregated features. However, this approach loses critical structural information inherent in the multigraph by failing to preserve individual edge features, making it unsuitable for tasks such as edge classification.

Additionally, since ADAMM does not compute latent features for the original edges, it cannot perform multi-edge aggregations repeatedly across multiple message passing layers. Another limitation of ADAMM is its lack of support for bi-directional message passing beyond merely incorporating edge direction as a feature. Previous works have shown that explicit bi-directional message passing improve accuracy for directed multigraphs Egressy et al. (2024).

**DIAM** Ding et al. (2024) models cryptocurrency transaction networks as directed, edge-attributed multigraphs and focuses on learning node representations that capture both temporal transaction patterns and structural discrepancies between illicit and benign accounts. The method introduces Edge2Seq, which constructs sequences from a node's incoming and outgoing edges separately and encodes them with GRUs to produce node-level embeddings. Although this sequence-based processing yields informative node-level representations, it does not update edge representations, as all edge information is immediately merged into node embeddings. DIAM further employs a Multigraph Discrepancy (MGD) module that performs directed message passing using both neighbor features and their differences from the target node, a design meant to emphasize behavioral deviations of illicit nodes rather than rely on homophily.

Despite achieving strong results on standard node classification benchmarks like ETH (see Table 1), the original DIAM model only computes node embeddings and does not support edge classification. However, we have extended it to include an edge classification head that leverages DIAM's node embeddings and the original edge features, which enabled us to apply DIAM also to AML datasets (Altman et al., 2023) which we include the results to Appendix E. The results in Table 15 show that on the AML task, DIAM outperformed by our method (MEGA-PNA), clearly reflecting the importance of our edge-attributed multigraph modeling approach in effectively capturing complex transaction patterns.

## B PROOFS

### B.1 PROOF OF THEOREM 1

For any function $f : \mathcal{X} \to \mathcal{Y}$, we write $\mathrm{Im}(f)$ to denote its image, i.e., the set $\{f(x) \mid x \in \mathcal{X}\} \subseteq \mathcal{Y}$.

*Proof.* We assume that each edge feature vector is equipped with a constant, i.e., $1 \in \mathbb{R}$. Hence, we can note that by using $g_1$, the model can compute the cardinality of the multiset.

We prove the claim by establishing two parts: (i) $\mathrm{Im}(\mathcal{T}_{\text{single-stage}}) \subseteq \mathrm{Im}(\mathcal{T}_{\text{two-stage}})$, (ii) $\mathrm{Im}(\mathcal{T}_{\text{single-stage}}) \neq \mathrm{Im}(\mathcal{T}_{\text{two-stage}})$.

**Part (i)** As stated in the Definition 3, $\mathcal{T}_{\text{single-stage}}$ aggregates all edge features in the neighborhood $\mathcal{X}_j$, treating the neighborhood as a single multiset,

$$\mathcal{X}_{\text{flat}} := \biguplus_{X_{ij} \in \mathcal{X}_j} X_{ij}.$$

Then the output of the single-stage aggregation is given by,

$$\mathcal{T}_{\text{single-stage}}(\mathcal{X}_j) = f_\theta(\mathcal{X}_{\text{flat}}) = \mathrm{MLP}_\theta\left([g_1(\mathcal{X}_{\text{flat}}) \,\|\, \ldots \,\|\, g_k(\mathcal{X}_{\text{flat}})]\right).$$

Two-stage aggregation scheme first applies $f_{\theta_1}$ to each multiset $X_{ij}$ *separately*, then $f_{\theta_2}$ is applied to resulting multiset of vectors in the second stage:

$$f_{\theta_1}(X_{ij}) = \mathrm{MLP}_{\theta_1}\left([g_1(X_{ij}) \,\|\, \ldots \,\|\, g_k(X_{ij})]\right),$$

and

$$\mathcal{T}_{\text{two-stage}}(\mathcal{X}_j) = f_{\theta_2}\left(\{\{f_{\theta_1}(X_{ij}) \mid X_{ij} \in \mathcal{X}_j\}\}\right).$$

Now we show that $\mathcal{T}_{\text{two-stage}}$ can compute what $\mathcal{T}_{\text{single-stage}}$ can compute.

We first consider the case $r = 1$, corresponding to the sum aggregator:

$$g_1(\mathcal{X}_{\text{flat}}) = \sum_{i \in N_{\text{in}}(j)} g_1(X_{ij})$$

Thus, by applying $g_1$ in both stages, the model can compute $g_1(\mathcal{X}_{\text{flat}})$.

Now for $2 \leq r \leq k$, the raw moments of the flattened multiset $\mathcal{X}_{\text{flat}}$, can be expressed as a weighted average of the raw moments of the individual multisets $X_{ij}$:

$$g_r\left(\mathcal{X}_{\text{flat}}\right) = \frac{1}{n} \sum_{X_{ij} \in \mathcal{X}_j} P_{ij} \cdot g_r(X_{ij}), \quad 2 \leq r \leq k,$$

where $n := \sum_i P_{ij}$ is the total number of edges.

Since each edge feature is equipped with a constant 1, the cardinality $P_{ij}=|X_{ij}|$ can be computed using the sum aggregator $g_1$. Thus, $f_{\theta_1}$ can compute $P_{ij} \cdot g_r(X_{ij})$ for each $r$.

In the second stage, the multiset $\{\{f_{\theta_1}(X_{ij}) \mid X_{ij} \in \mathcal{X}_j\}\}$ contains all such terms, and the sum aggregator $g_1$ in $f_{\theta_2}$ can compute

$$\sum_{X_{ij} \in \mathcal{X}_j} P_{ij} \cdot g_r(X_{ij}), \quad n = \sum_i P_{ij}$$

thus enabling $f_{\theta_2}$ to compute the raw moments $g_r(\mathcal{X}_{\text{flat}})$.

Hence, we conclude that:

$$\text{Im}(\mathcal{T}_{\text{single-stage}}) \subseteq \text{Im}(\mathcal{T}_{\text{two-stage}}).$$

**Part (ii):** We now show that $\text{Im}(\mathcal{T}_{\text{single-stage}}) \neq \text{Im}(\mathcal{T}_{\text{two-stage}})$.

We begin by defining a simple function that cannot be computed by any single-stage aggregation scheme.

$$F_r(\mathcal{X}_j) := \sum_{X_{ij} \in \mathcal{X}_j} g_r(X_{ij}), \quad 2 \leq r \leq k$$

Such functions, $F_r$, are computable by $\mathcal{T}_{\text{two-stage}}$ since by design it preserves the partitioning over distinct neighbors of node $j$, allowing $f_{\theta_2}$ to operate on a multiset of per-neighbor representations.

In contrast, any function computed by $\mathcal{T}_{\text{single-stage}}$ is in the form:

$$\mathcal{T}_{\text{single-stage}}(\mathcal{X}_j) = f_\theta(\mathcal{X}_{\text{flat}}) = \text{MLP}_\theta\left([g_1(\mathcal{X}_{\text{flat}}) \| \ldots \| g_k(\mathcal{X}_{\text{flat}})]\right).$$

Let $n := |\mathcal{X}_{\text{flat}}|$, we now analyze two distinct cases:

**Case 1:** $k \geq n$   When $k \geq n$, the number of aggregators in $f_\theta$ is sufficient to reconstruct the entire neighborhood multiset $\mathcal{X}_{\text{flat}}$ without loss of information, as established in Theorem 1 of Corso et al. (2020). However, since $\mathcal{X}_{\text{flat}}$ does not contain any information about which neighbor $i$ each edge originates from, even a full reconstruction of the neighborhood does not allow the recovery of per-neighbor partitions. Therefore, the function $F_r$ is incomputable by $\mathcal{T}_{\text{single-stage}}$.

**Case 2:** $k < n$   In this case $\mathcal{T}_{\text{single-stage}}$ cannot even reconstruct the full multiset $\mathcal{X}_{\text{flat}}$, as the number of aggregators $k$ is insufficient to discriminate between multisets of size $n$, as stated in Theorem 1 in Corso et al. (2020). As a result, the function $F_r$ remains incomputable by $\mathcal{T}_{\text{single-stage}}$.

Therefore, we have proven that two-stage aggregation induces a strictly larger image than single-stage aggregation,

$$\text{Im}(\mathcal{T}_{\text{single-stage}}) \subsetneq \text{Im}(\mathcal{T}_{\text{two-stage}}).$$

$\square$

### B.2 PROOF OF PROPOSITION 2

For consistency we use the same notation introduced in Section 3.1. Let $\mathcal{G} = (\mathcal{V}, \mathcal{E})$ be a multigraph with node features $\mathbf{x}_i \in \mathbb{R}^D$ and edge features $\mathbf{e}_{ijp} \in \mathbb{R}^K$. We assume that each edge carries a distinct feature vector. Each edge $e \in \mathcal{E}$ is assigned a port number $\rho(e)$ by a given port numbering scheme, and these port numbers are incorporated into the edge features, as proposed by Egressy

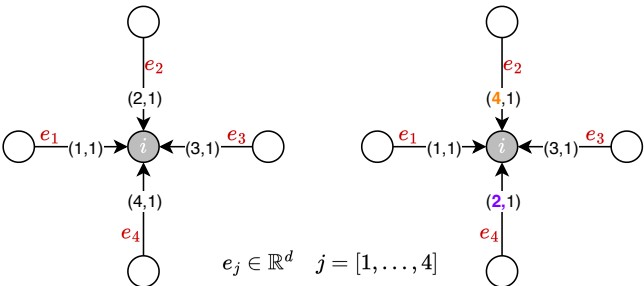

Figure 8: Illustration of counter example for the permutation equivariance of Multi-GNN Egressy et al. (2024). The left panel shows the graph $\mathcal{G}$ with one permutation of the port numbering, while the right panel illustrates a different permutation of the assigned port numbers. Assume each edge has distinct features, i.e $e_2 \neq e_4$.

et al. (2024). We further assume that the initial edge feature vectors $\mathbf{e}_{ijp}^{(0)} < \mathbf{e}_{i'j'p'}^{(0)}$, induce a strict total order over the edges, which permits a deterministic port assignment.

Now consider a scenario where there is no strict total ordering of the edges. As a result, the assignment of port numbers is arbitrary. For a node $i$ with $d$ incoming edges, there are $d!$ possible port numbering, corresponding to all possible permutations.

The latent feature of the node $i$ at layer $l$ is computed as:

$$\mathbf{x}_i^{(l)} = \sum_{j \in N(i)} \sum_{p \in P_{ji}} \phi\big(\mathbf{x}_j^{(l-1)}, [\mathbf{e}_{jip}^{(l-1)} \| \rho(j,i)]\big). \tag{12}$$

where $\phi$ is the message function, and $\rho(j,i) \in \mathbb{R}^2$ is the port numbers assigned to the edge between node $j$ and node $i$, as shown in Figure A.2(a)

We proceed by proof by contradiction. Suppose that the GNN with port numbering is permutation equivariant at the graph level, that is, permuting the node and edge indices results in an equivalent permutation of the output given as in Definition 5. This property requires the model to be permutation invariant over each node's neighborhood: reordering the incoming edges (i.e., permuting the port numbers) should not affect a node's representation.

Let $\sigma$ be a permutation of the port numbers. Applying this permutation to the port numbers of the edges yields a new port assignment $\rho_\sigma(e)$. The updated representation of node $i$ at layer $l$ under this permuted port assignment is:

$$\hat{\mathbf{x}}_i^{(l)} = \sum_{j \in N(i)} \sum_{p \in P_{ji}} \phi\big(\mathbf{x}_j^{(l-1)}, [\mathbf{e}_{jip}^{(l-1)} \| \rho_\sigma(j,i)]\big). \tag{13}$$

By assumption:

$$\mathbf{x}_i^{(l)}(\rho) = \hat{\mathbf{x}}_i^{(l)}(\rho_\sigma) \tag{14}$$

However, since $\phi$ explicitly depends on the port number $\rho(j,i)$, permuting the port numbers alters the input to $\phi$ via concatenated feature $[\mathbf{e}_{jip}^{(l-1)} \| \rho_\sigma(j,i)]$. Assuming that each edge carries a distinct feature vector, as is typical in settings like financial transaction networks, this change affects the resulting messages and, consequently, the updated representation of node $i$. Hence permuting assigned port numbers leads to $\mathbf{x}_i^{(l)}(\rho) \neq \hat{\mathbf{x}}_i^{(l)}(\rho_\sigma)$, violating permutation invariance over the neighborhood of $i$, contradicting the assumption of GNN being permutation equivariant.

Figure 8 illustrates this contradiction: node $i$ has four incoming edges, and permuting their port numbers leads to different messages and a different update. We thus conclude that arbitrary port numbering breaks permutation equivariance.

### B.3 Proof of Proposition 1

We adopt the notation and terminology introduced in Section 3.1 of this paper to ensure consistency and ease of reference.

*Proof:* The proposed message passing layer performs two aggregations over the neighborhood of a target node $j$. The first is the multi-edge aggregation, in which the latent features of the multi-edges are aggregated at artificial nodes. The multiset of such features are denoted as,

$$X_{ij} = \{\{\mathbf{e}_{ijp} \mid p \in [P_{ij}]\}\} \tag{15}$$

The vectors in the multiset $X_{ij}$ are aggregated on artificial nodes,

$$\mathbf{h}_{ij} = f_{\theta_1}(X_{ij}). \tag{16}$$

Since aggregators in $f_{\theta_1}$ are assumed to be permutation invariant, for any permutation function $\rho$ acting on multi-edges, we have $f_{\theta_1}(\rho \cdot X_{ij}) = f_{\theta_1}(X_{ij})$.

The second aggregation is then performed over the neighborhood of the target nodes, all of which happen to be artificial nodes associated with distinct neighbors in the original graph (see Figure 2).

$$H_{N_{in}(j)} = \{\{\mathbf{h}_{ij} \mid (i,j) \in N_{in}(j))\}\} \in \mathcal{M}_d. \tag{17}$$

Hence, the second aggregation operates over the multiset $H_{N_{in}(j)}$,

$$\mathbf{x}_j = f_{\theta_2}(H_{N_{in}(j)}) \in \mathcal{M}_d. \tag{18}$$

Again since the aggregators in $f_{\theta_2}$ are assumed to be permutation invariant for any permutation function $\pi$ acting on the neighbors of a target node $j$, we have $f_{\theta_2}(\pi \cdot H_{N_{in}(j)}) = f_{\theta_2}(H_{N_{in}(j)})$.

Our framework MEGA-GNN, integrates the two-stage aggregation scheme (as defined in Definition 4) using aggregation functions $f_{\theta_1}$ and $f_{\theta_2}$ within a single message passing layer, as detailed in Section 3.3. Since the composition of permutation invariant functions remains permutation invariant, our message passing layer ($f_{\theta_1} \circ f_{\theta_2}$) is invariant to the permutations of neighboring nodes and edges. Unlike simple graphs, node permutations do not directly imply edge permutations in multigraphs due to the presence of multi-edges. Thus, we explicitly define the permutation of multi-edges, $\rho$, ensuring that our message passing layer remains permutation-invariant to both nodes and edges in the neighborhood of the target node.

Finally, as demonstrated by Bronstein et al. (2021), the composition of permutation invariant layers ($f = f_{\theta_1} \circ f_{\theta_2} \circ f_{\theta_1} \circ f_{\theta_2} \cdots$) allows the construction of functions $f$ that are equivariant to symmetry group actions. In the multigraph domain, this symmetry group includes permutations of both nodes and edges. The overall permutation equivariance of the MEGA-GNN model follows from the fact that each permutation invariant message passing layer operates independently on each node's neighborhood, regardless of the ordering of nodes or edges. Specifically, for any permutation $g \in \sum_n$ acting on the set of node and edges, the model's output satisfies $f(g \cdot X) = g \cdot f(X)$.

### B.4 Proof of Theorem 2

*Proof:* Given a connected directed multigraph $G(V, E)$ with $n$ nodes and $m$ edges, assume that there exists an injective edge-labeling function $L : \mathcal{E} \to [M]$ for some $M \geq m$ that assigns a unique labels to each edge. We will prove that MEGA-GNN can compute unique node IDs under these assumptions.

Egressy et al. (2024) showed that a GNN can mimic a Breadth-First Search (BFS) algorithm to compute unique node IDs given pre-computed port numbers for the edges. We follow the same BFS-based approach and derive unique node ids without relying on pre-computed port numbers. Instead of the pre-computed port numbers, we use the unique edge labels provided by $L(e)$ to guide the node ID assignment process. As in Egressy et al. (2024), we use the Universal Approximation Theorem Hornik et al. (1989) for MLPs, to avoid explicit construction of the MEGA-GNN layers. We also assume that the MEGA-GNN aggregates the multi-edges by computing their minimum, which is followed by a node-level aggregation, where an MLP is applied element-wise to the incoming messages, followed by another minimum computation.

---

**Algorithm 1** BFS Node ID Assignment

---

**Input:** Connected directed multigraph $G = (V, E)$ with $n$ nodes and $m$ edges, diameter $D$, and root node $r \in V$. Active nodes $X \subseteq V$ and finished nodes $F \subseteq V$. Edge Labeling $L : E \to [M]$ for some $M \geq m$.

**Output:** Unique node IDs $h(v)$ for all $v \in V$ (in base $2n$)

1: $h(r) \leftarrow 1$;    $h(v) \leftarrow 0$ for all $v \in V \setminus \{r\}$
2: $F \leftarrow \emptyset$;    $X \leftarrow \{r\}$
3: **for** $k \leftarrow 1$ **to** $D$ **do**
4:    **for** $v \in V$ **do**
5:      **if** $v \in X$ **then**
6:        send $h(v) \parallel \min\{L((v,u))_{\text{out}}\}$ to $u \in N_{\text{out}}(v)$
7:        send $h(v) \parallel M + \min\{L((u,v))_{\text{in}}\}$ to $u \in N_{\text{in}}(v)$
8:        $F \leftarrow F \cup \{v\}$;    $X \leftarrow X \setminus \{v\}$
9:      **end if**
10:      **if** $v \notin F$ **then**
11:        **if** Incoming messages $M(v) \neq \emptyset$ **then**
12:          $h(v) \leftarrow \min\{M(v)\}$
13:          $X \leftarrow X \cup \{v\}$
14:        **end if**
15:      **end if**
16:    **end for**
17: **end for**

---

Following the approach of Egressy et al. (2024), the node ID assignment algorithm starts from a root node (also called the ego node) and assigns IDs to all the other nodes connected to it via message passing . We are not going to reiterate the setup of the entire proof and focus on the differences. Specifically, instead of pre-computed port numbers, we assume the existence of a injective edge-labeling function, $L : E \to [M]$, which is induced by the original edge features. This injective mapping naturally defines a strict total order over the edges, allowing MEGA-GNN to deterministically select, among multi-edges, the edge with the minimum label during aggregation. This ensures a consistent and unique node ID assignment while relying solely on intrinsic edge features.

Our MEGA-GNN model, which mimics Algorithm 1, assigns ids to each node connected to the root node. What remains to be shown is that those assigned node IDs are unique. First, note that nodes at different distances from the root cannot end up with the same node ID. A node at distance $k$ will receive its first proposal in round $k$ and, therefore, it will have an ID with exactly $k + 1$ digits. Furthermore, an inductive argument shows that active nodes (nodes at the same distance) cannot have the same node IDs. Certainly, the argument is also true at the start when $X = \{r\}$. Now assuming all active nodes from the previous round $(k - 1)$ had distinct node IDs, then the only way two active nodes (in round $k$) can have the same ID is if they accept a proposal from the same neighboring node. This is because, based on the induction hypothesis, proposals from different nodes will already differ in their first $k - 1$ digits. If two active nodes accepted a proposal from the same node, then they would have received different edge labels, a strict total ordering among the edges enables assignment of distinct edge labels. In addition, because $m$ is added to all incoming labels, incoming labels cannot be the same as the outgoing labels. Therefore the active nodes always accept unique proposals.

### B.5 Proof of Theorem 3

*Proof.* Let $\mathcal{V}$ denote the set of nodes, $\mathcal{E}$ the multiset of edges, and $\mathcal{E}^{\text{supp}} \subseteq \mathcal{E}$ the set of unique $(i, j)$ pairs with multiplicity at least one. Let $d$ be the dimensionality of node and edge embeddings, and assume all linear transformations map $\mathbb{R}^d \to \mathbb{R}^d$.

A single MEGA-GNN layer executes two-stage aggregation. In each stage, the three terms below correspond to costs of aggregation, linear layer, and nonlinearity; the term is the cost of the edge update. The per-layer cost is

$$\mathcal{O}\big( \underbrace{|\mathcal{E}|\, d + |\mathcal{E}^{\text{supp}}|\, d^2 + |\mathcal{E}^{\text{supp}}|\, d}_{\text{first aggregation stage}} + \underbrace{|\mathcal{E}^{\text{supp}}|\, d + |\mathcal{V}|\, d^2 + |\mathcal{V}|\, d}_{\text{second aggregation stage}} + \underbrace{|\mathcal{E}|\, d^2 + |\mathcal{E}| d}_{\text{edge update with non-linearity}} \big).$$

Since $|\mathcal{E}^{\text{supp}}| \leq |\mathcal{E}|$, the total complexity simplifies to

$$\mathcal{O}\big(\,(|\mathcal{E}| + |\mathcal{V}|)\,d^2 + (|\mathcal{E}| + |\mathcal{V}|)\,d\,\big).$$

□

For comparison, a standard message-passing GNN with edge updates (per layer) has

$$\mathcal{O}\big(\underbrace{|\mathcal{E}|\,d}_{\text{neighborhood agg.}} + \underbrace{|\mathcal{V}|\,d^2}_{\text{node update}} + \underbrace{|\mathcal{V}|\,d}_{\text{nonlinearity}} + \underbrace{|\mathcal{E}|\,d^2}_{\text{edge update}} + \underbrace{|\mathcal{E}|\,d}_{\text{nonlinearity}}\big)$$

which sums to $\mathcal{O}\big(\,(|\mathcal{E}| + |\mathcal{V}|)d^2 + (|\mathcal{E}| + |\mathcal{V}|)d\,\big)$.

This shows that MEGA-GNN has the same asymptotic complexity as standard message-passing GNNs that perform edge updates.

## C  IMPLEMENTATION DETAILS

### C.1  ARCHITECTURE DIAGRAMS

Figure 3 illustrate a single layer of the MEGA-GNN architecture. Figure 9 shows the MEGA-GNN layer equipped with bi-directional message-passing capabilities. The figures use the same notation as in Sections 3.3 and 3.4 for clarity.

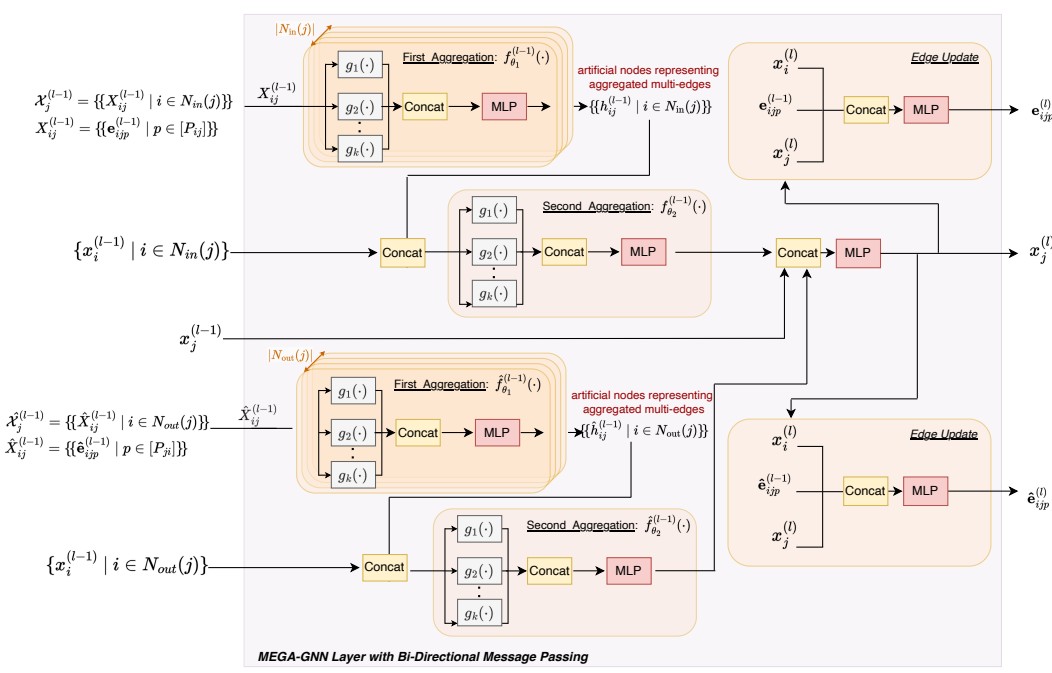

Figure 9: Overview of the MEGA-GNN layer with a bi-directional message passing. In directed multigraphs, reverse edges are added opposite to the original edges. Separate message computations are performed for original and reversed edges. The diagram illustrates the message passing scheme described in Sections 3.3 and 3.4, using consistent notation.

### C.2  COMPARISON POINTS

Table 7 summarizes the baseline methods and multigraph adaptations evaluated in our experiments.

Table 7: Backbone architectures and multigraph adaptations: checkmarks indicate which combinations are evaluated in our experiments.

| Multigraph Adaptations | GIN | PNA | GenAgg | R-GCN | FraudGT |
|---|---|---|---|---|---|
| Base (no adaptations) | ✓ | ✓ | ✓ | ✓ | ✓ |
| Multi Egressy et al. (2024) | ✓ | ✓ | ✓ | | ✓ |
| ADAMM Sotiropoulos et al. (2023) | ✓ | ✓ | ✓ | | |
| MEGA (ours) | ✓ | ✓ | ✓ | ✓ | |

Table 8: Hyperparameter settings for AML and ETH datasets

| | GIN | | PNA | | R-GCNE |
|---|---|---|---|---|---|
| | AML | ETH | AML | ETH | AML |
| *lr* | 0.003 | 0.006 | 0.0008 | 0.0008 | 0.003 |
| *hidden_dim* | 64 | 32 | 20 | 20 | 32 |
| *batch_size* | 8192 | 4096 | 8192 | 4096 | 8192 |
| *dropout* | 0.1 | 0.1 | 0.28 | 0.1 | 0.1 |
| *w_ce1, w_ce1* | 1, 6.27 | 1, 6.27 | 1, 7 | 1, 3 | 1, 6.27 |

## C.3 HYPERPARAMETERS

For each base GNN model and dataset, we utilized a distinct set of hyperparameters, as detailed in Table 8. The MEGA-GenAgg and Multi-GenAgg models employed the aggregation function proposed by Kortvelesy et al. (2023). In all experiments involving GenAgg, we adopted the default layer sizes of $(1, 2, 2, 4)$, and both the $a$ and $b$ parameters were made learnable, allowing the model to tailor the aggregation function to the specific downstream task. Additionally, for the GenAgg experiments, we applied the hyperparameters configured for GIN-based models as shown in Table 8.

For the AML dataset, the model was operated on neighborhoods constructed around the seed edges, while for the ETH dataset, the neighborhoods were selected around the seed nodes. In both datasets, we sampled 2-hop neighborhoods, selecting 100 neighbors per hop.

## C.4 AML AND ETH DATASETS

Table 9 provides an overview of the datasets used in our experiments. The AML dataset is available in three different scales (Small, Medium, and Large), each with High Illicit (HI) and Low Illicit (LI) versions.

Table 9: Statistics of AML (Altman et al., 2023), ETH (Chen et al., 2021a) and JODIE (Kumar et al., 2019) datasets.

| Dataset | # Nodes | # Edges | Illicit Rate | Split [%] |
|---|---|---|---|---|
| AML Small HI | 515,088 | 5,078,345 | 0.102% | 64/19/17 |
| AML Small LI | 705,907 | 6,924,049 | 0.051% | 64/19/17 |
| AML Medium HI | 2,077,023 | 31,898,238 | 0.110% | 61/17/22 |
| AML Medium LI | 2,032,095 | 31,251,483 | 0.051% | 61/17/22 |
| AML Large HI | 2,116,168 | 179,702,229 | 0.124% | 60/20/20 |
| AML Large LI | 2,070,980 | 176,066,557 | 0.057% | 60/20/20 |
| ETH | 2,973,489 | 13,551,303 | 0.04% | 65/15/20 |
| MOOC | 7,144 | 411,749 | 0.98% | 60/20/20 |
| Reddit | 10,984 | 672,447 | 0.05% | 60/20/20 |
| Wikipedi | 9,227 | 157,474 | 0.14% | 60/20/20 |

**AML Data Split:** We adopt the same temporal splitting strategy as proposed in Egressy et al. (2024), which follows train-validation-test split based on transaction timestamps. Specifically, we

sort all transactions and define two cut-off points, $t_1$ and $t_2$, to partition the data as summarized in Table 9. Transactions occurring before $t_1$ are used for training, those between $t_1$ and $t_2$ for validation, and those after $t_2$ for testing. Since validation and test transactions may depend on patterns in earlier activity, we construct three dynamic graph snapshots at times $t_1$, $t_2$, and $t_3 = t_{\max}$, the latest timestamp in the dataset. The train graph includes only training transactions and their corresponding nodes. The validation graph includes both training and validation transactions but computes metrics only on the validation indices. Similarly, the test graph contains all transactions in the given dataset, with evaluation performed solely on the test indices. This dynamic setup mirrors real-world usage in financial institutions, where systems must detect anomalies in new batches of transactions while leveraging historical context.

**ETH Data Split:** Similar to AML we use a temporal train-validation-test split. We order the nodes by the first transaction they are involved in (either as sender or receiver) before splitting. Again this gives us threshold times t1 and t2, and we use these times to create our train, validation, and test graphs.

## C.5 MEGA VARIANTS

In this section, we provide detailed descriptions of the MEGA-GIN, MEGA-PNA, MEGA-GenAgg, and MEGA-RGCNE models used in our study.

As introduced in Sections 3.1 and 3.2, our framework employs a two-stage aggregation scheme with two aggregation functions, $f_{\theta_1}$ and $f_{\theta_2}$, each constructed from a set of $k$ aggregators $g_1, \ldots, g_k :$ $\mathcal{M}_d \to \mathbb{R}^d$. Specifically, we define:

$$f_\theta(X) := \mathrm{MLP}_\theta\big([g_1(X) \,\|\, \ldots \,\|\, g_k(X)]\big), \quad f_\theta : \mathcal{M}_d \to \mathbb{R}^{d'}. \tag{19}$$

The MEGA variants differ in the choice and number of aggregators used in the two-stage process.

**MEGA-GIN.** Following the Graph Isomorphism Network (GIN) model proposed by Xu et al.Xu et al. (2019), MEGA-GIN uses a single aggregator, namely SUM, in both $f_{\theta_1}$ and $f_{\theta_2}$. That is, $k = 1$ and $g_1 = \mathrm{SUM}$.

**MEGA-PNA.** The MEGA-PNA builds on the Principal Neighbourhood Aggregation (PNA) framework proposed by Corso et al. Corso et al. (2020), which combines multiple statistical aggregators. Accordingly, we use $k = 4$ aggregators: MEAN, MAX, MIN, and STD, applied in both $f_{\theta_1}$ and $f_{\theta_2}$.

**MEGA-GenAgg.** The MEGA-GenAgg employs a single, learnable aggregator as introduced in the GenAgg framework Kortvelesy et al. (2023). Unlike fixed statistical functions, the aggregator in GenAgg is parameterized and trained end-to-end. Accordingly, we set $k = 1$, and use $g_1 = \mathrm{GENAGG}$ in both $f_{\theta_1}$ and $f_{\theta_2}$.

**MEGA-RGCNE.** The MEGA-RGCNE variant integrates the expressive multi-aggregator scheme of PNA Corso et al. (2020) in the first aggregation stage $f_{\theta_1}$, where we use $k = 4$ aggregators: MEAN, MAX, MIN, and STD. In the second stage $f_{\theta_2}$, we incorporate relation-specific transformation matrices $W_r^{(l)}$, applying distinct linear transformations for each relation type $r$, as shown in Figure 6. This design demonstrates the flexibility of the MEGA framework, which allows different aggregation strategies to be combined across stages.

## C.6 EDGE-AUGMENTED R-GCN (R-GCNE)

To incorporate edge attributes, we extend the standard Relational Graph Convolutional Network (R-GCN) Schlichtkrull et al. (2018) by introducing the Relational Graph Convolutional Network with Edge features (R-GCNE). In this variant, edge features are also included in the message passing formulation.

Consider the notation introduced in Section 3.1. Building on that, we define the set of relation types $\mathcal{R}$, where each edge $(j, r, i) \in \mathcal{E}$ is labeled with a relation $r \in \mathcal{R}$. Each relation type $r$ is

associated with a learnable transformation matrix $W_r^{(l)} \in \mathbb{R}^{d \times d}$, and we define $W_0^{(l)} \in \mathbb{R}^{d \times d}$ as the transformation matrix applied to a node's own features (i.e., self-loop) at layer $l$. For each node $i \in \mathcal{V}$, we define the relation-specific neighborhood as $N_i^r := \{j \in \mathcal{V} \mid (j, r, i) \in \mathcal{E}\}$, representing the set of incoming neighbors connected via relation $r$.

The message passing equations for both models are defined as follows:

- **R-GCN Message Passing Equation:**

$$x_i^{(l+1)} := \sigma \left( x_i^{(l)} W_0^{(l)} + \sum_{r \in \mathcal{R}} \sum_{j \in N_i^r} \frac{1}{|N_i^r|} x_j^{(l)} W_r^{(l)} \right), \quad x_i^{(l+1)} \in \mathbb{R}^d.$$

- **R-GCNE Message Passing Equation:**

$$x_i^{(l+1)} := \sigma \left( x_i^{(l)} W_0^{(l)} + \sum_{r \in \mathcal{R}} \sum_{j \in N_i^r} \frac{1}{|N_i^r|} \left( x_j^{(l)} W_r^{(l)} + e_{ji}^{(l)} W_r^{(l)} \right) \right), \quad x_i^{(l+1)} \in \mathbb{R}^d.$$

If R-GCNE is applied to multigraphs with edge relations, we must also account for multiple edges between the same pair of nodes with the same relation. To handle this, similar to Section 3.1, we define the multiset

$$X_{ij,r} = \{\{\mathbf{e}_{ijp,r} \mid p \in [P_{ij,r}]\}\}$$

denote the multiset of edge feature vectors from node $j$ to node $i$ under relation $r$, where $P_{ij,r}$ is the number of such edges. We denote the feature of the $p$-th such edge as $e_{jip,r}^{(l)} \in \mathbb{R}^d$.

Then the formulation of R-GCNE on multigraphs with relation types becomes:

$$x_i^{(l+1)} := \sigma \left( x_i^{(l)} W_0^{(l)} + \sum_{r \in R} \sum_{j \in N_i^r} \sum_{p=1}^{P_{jri}} \frac{1}{|N_i^r|} (x_j^{(l)} W_r^{(l)} + e_{jip,r}^{(l)} W_r^{(l)}) \right) \in \mathbb{R}^d.$$

# D  COMPUTATION AND MEMORY COSTS

## D.1  COMPUTATION COSTS

Table 10: Training time (seconds per epoch) and memory consumption on the AML Small HI dataset.

| Model | # Params | Train sec/ep | Mem. Usage (GB) |
|---|---|---|---|
| GIN | 69.6K | 2.57 | 2.1 |
| PNA | 32.2K | 8.19 | 5.09 |
| GenAgg | 69.7K | 6.90 | 4.25 |
| FraudGT | 182.4K | 28.63 | 11.59 |
| R-GCNE | 47.7K | 11.67 | 13.78 |
| Multi-GIN | 128.3K | 9.41 | 7.47 |
| Multi-PNA | 60.0K | 26.46 | 21.08 |
| Multi-GenAgg | 128.4K | 25.56 | 13.34 |
| Multi-FraudGT | 243.7K | 85.85 | 18.2 |
| MEGA-GIN | 161.3K | 9.08 | 8.47 |
| MEGA-PNA | 79.2K | 28.58 | 19.73 |
| MEGA-GenAgg | 128.4K | 30.55 | 16.22 |
| MEGA-R-GCNE | 138.3K | 19.18 | 8.58 |

Table 10 reports the training time per epoch (in seconds), memory usage, and number of parameters for all evaluated models on the AML Small HI dataset, measured on the same hardware. While our models were trained on an AI cluster, the results reported in this table were measured on a single NVIDIA GeForce RTX 4090 GPU to ensure fair and consistent comparison across all models. The reported training-time values were measured over 256 iterations per epoch, and the memory usage

represents the maximum GPU utilization. Table 11 reports the inference throughput rate in transactions per seconds for AML Small HI, AML Small LI, and ETH datasets, measured on the same NVIDIA GeForce RTX 4090 GPU. These results demonstrate that MEGA-GNNs are only slightly slower than Multi-GNNs, confirming that the overhead of two-state aggregation is not significant.

Table 11: Inference performance: throughput rate in [trans/sec].

| Model | AML Small HI | AML Small LI | ETH |
|---|---|---|---|
| Multi-GIN | 30179 | 26945 | 222825 |
| MEGA-GIN | 28655 | 25617 | 197236 |
| Multi-PNA | 28389 | 21153 | 123692 |
| MEGA-PNA | 27012 | 20344 | 117354 |

### D.2 MEMORY OVERHEAD

In terms of memory, we assume that the multigraph is attributed, with feature vectors associated with each edge. The memory required to store the edge embeddings is $O(|\mathcal{E}|d)$ in the standard case.

For the two-stage aggregation, additional memory is needed to store a tensor of size $|\mathcal{E}|$ that indexes parallel edges in the multiset. This tensor is computed during preprocessing and reused across batches, thus avoiding redundant calculations. Furthermore, a tensor of size $|\mathcal{E}^{supp}| \times d$ is created dynamically during the forward pass to store the features of the artificial nodes; there are temporary computational intermediates during the first aggregation stage, similar to how hidden states function within neural network layers. These nodes do not permanently expand the graph structure.

As a result, the additional memory overhead per batch is $|\mathcal{E}| + |\mathcal{E}^{supp}|d$, which is $O(|\mathcal{E}|d)$. Consequently, the two-stage aggregation does not asymptotically alter the overall memory complexity, meaning that the model remains efficient even as the size of the input graph increases.

## E  AML RESULTS AND ADDITIONAL PERFORMANCE METRICS

Table 12: Minority class F1 scores (%) for six AML datasets using different GNN baselines (GIN,PNA, GenAgg, FraudGT and R-GCNE) and multigraph adaptations (Multi and MEGA). We extended R-GCNE to support only MEGA adaptations. Furthermore, we were unable to obtain Multi-GenAgg and MEGA-GenAgg results for the Large datasets.

| Model | Small HI | Small LI | Medium HI | Medium LI | Large HI | Large LI |
|---|---|---|---|---|---|---|
| GIN | $46.50 \pm 4.11$ | $19.93 \pm 3.55$ | $58.65 \pm 2.50$ | $25.36 \pm 1.49$ | $49.80 \pm 1.38$ | $4.99 \pm 3.66$ |
| PNA | $62.96 \pm 1.43$ | $21.02 \pm 4.05$ | $66.87 \pm 1.87$ | $31.79 \pm 2.30$ | $55.01 \pm 1.94$ | $20.47 \pm 1.93$ |
| GenAgg | $56.45 \pm 2.94$ | $21.03 \pm 2.23$ | $54.21 \pm 7.90$ | $20.72 \pm 2.60$ | $52.23 \pm 4.29$ | $9.23 \pm 3.07$ |
| FraudGT | $69.68 \pm 1.58$ | $28.69 \pm 2.05$ | $63.38 \pm 0.87$ | $24.02 \pm 0.52$ | $54.35 \pm 1.65$ | $11.02 \pm 2.65$ |
| R-GCNE | $63.91 \pm 3.18$ | $37.40 \pm 1.61$ | $65.71 \pm 0.61$ | $35.70 \pm 0.99$ | $58.26 \pm 1.08$ | $23.32 \pm 0.73$ |
| GFP+LightGBM | $62.86 \pm 0.25$ | $20.83 \pm 1.50$ | $59.48 \pm 0.15$ | $20.85 \pm 0.38$ | $48.67 \pm 0.24$ | $17.09 \pm 0.46$ |
| GFP+XGBoost | $63.23 \pm 0.17$ | $27.30 \pm 0.33$ | $65.70 \pm 0.26$ | $28.16 \pm 0.14$ | $42.68 \pm 12.93$ | $24.23 \pm 0.12$ |
| Multi-FraudGT | $\mathbf{75.81 \pm 0.75}$ | $\underline{45.69 \pm 1.14}$ | $75.97 \pm 0.18$ | $44.66 \pm 0.58$ | $\underline{73.04 \pm 0.59}$ | $\underline{35.49 \pm 0.52}$ |
| Multi-GIN | $62.66 \pm 1.73$ | $32.21 \pm 0.99$ | $67.72 \pm 0.94$ | $31.24 \pm 2.12$ | $71.44 \pm 1.25$ | $9.46 \pm 8.85$ |
| Multi-PNA | $67.35 \pm 2.89$ | $35.39 \pm 3.93$ | $76.12 \pm 0.69$ | $43.81 \pm 0.51$ | $72.35 \pm 1.14$ | $33.54 \pm 2.04$ |
| Multi-GenAgg | $64.92 \pm 3.85$ | $36.36 \pm 4.07$ | $66.45 \pm 1.30$ | $37.72 \pm 0.73$ | OOM | OOM |
| MEGA-RGCNE | $70.65 \pm 1.80$ | $40.92 \pm 2.69$ | $74.48 \pm 0.25$ | $41.21 \pm 0.45$ | $67.41 \pm 3.38$ | $27.25 \pm 0.82$ |
| MEGA-GIN | $70.83 \pm 2.19$ | $43.67 \pm 0.55$ | $70.77 \pm 2.76$ | $39.03 \pm 1.88$ | $70.41 \pm 2.74$ | $11.64 \pm 1.64$ |
| MEGA-PNA | $74.01 \pm 1.55$ | $\mathbf{46.32 \pm 2.07}$ | $\mathbf{78.26 \pm 0.11}$ | $\mathbf{49.40 \pm 0.54}$ | $\mathbf{76.95 \pm 0.44}$ | $\mathbf{38.31 \pm 1.53}$ |
| MEGA-GenAgg | $\underline{74.48 \pm 0.84}$ | $46.30 \pm 0.42$ | $\underline{76.70 \pm 0.32}$ | $\underline{44.90 \pm 0.06}$ | OOM | OOM |

In this section, we present the results from Figure 4 in a tabular format with additional comparisons at Table 12. Additionally, we provide comprehensive evaluation results in Tables 13 and 14 which present Precision and Recall metrics for the AML edge classification task. These detailed metrics

Table 13: Precision scores (%) on AML edge classification task. Best result is indicated with **bold**.

| Model | Small HI | Small LI | Medium HI | Medium LI | Large HI | Large LI |
|---|---|---|---|---|---|---|
| GIN | $43.78 \pm 6.41$ | $17.90 \pm 4.92$ | $63.19 \pm 6.11$ | $29.00 \pm 2.45$ | $47.32 \pm 2.79$ | $32.07 \pm 22.91$ |
| PNA | $66.92 \pm 4.06$ | $20.47 \pm 6.80$ | $69.29 \pm 3.23$ | $49.01 \pm 4.75$ | $50.32 \pm 3.14$ | $46.19 \pm 7.38$ |
| GenAgg | $55.68 \pm 4.98$ | $22.55 \pm 9.02$ | $50.50 \pm 12.18$ | $23.01 \pm 4.95$ | $51.15 \pm 10.55$ | $35.16 \pm 10.81$ |
| RGCNE | $76.08 \pm 3.55$ | $\mathbf{68.90 \pm 3.86}$ | $75.37 \pm 2.64$ | $55.26 \pm 5.91$ | $72.43 \pm 3.93$ | $39.52 \pm 1.59$ |
| Multi-FraudGT | $\mathbf{80.04 \pm 1.36}$ | $68.07 \pm 3.34$ | $81.18 \pm 1.08$ | $73.24 \pm 1.42$ | $80.00 \pm 3.49$ | $63.38 \pm 3.39$ |
| Multi-GIN | $61.02 \pm 2.60$ | $33.61 \pm 3.44$ | $69.77 \pm 3.89$ | $36.43 \pm 6.98$ | $76.68 \pm 4.55$ | $47.78 \pm 33.73$ |
| Multi-PNA | $66.16 \pm 6.59$ | $43.99 \pm 8.72$ | $78.32 \pm 5.42$ | $67.22 \pm 3.31$ | $74.46 \pm 3.07$ | $\mathbf{72.68 \pm 6.25}$ |
| Multi-GenAgg | $64.66 \pm 5.54$ | $49.55 \pm 12.38$ | $67.45 \pm 0.78$ | $48.35 \pm 1.73$ | OOM | OOM |
| MEGA-RGCNE | $75.05 \pm 3.87$ | $58.49 \pm 12.59$ | $82.28 \pm 2.12$ | $74.75 \pm 2.86$ | $67.12 \pm 7.39$ | $69.26 \pm 11.00$ |
| MEGA-GIN | $70.11 \pm 4.23$ | $63.95 \pm 4.29$ | $74.16 \pm 4.91$ | $67.32 \pm 11.24$ | $70.35 \pm 6.59$ | $38.35 \pm 5.36$ |
| MEGA-PNA | $76.90 \pm 4.05$ | $66.26 \pm 8.82$ | $84.26 \pm 0.62$ | $\mathbf{75.74 \pm 2.75}$ | $\mathbf{83.81 \pm 1.27}$ | $57.28 \pm 8.92$ |
| MEGA-GenAgg | $78.27 \pm 2.46$ | $66.16 \pm 1.22$ | $\mathbf{85.57 \pm 1.29}$ | $72.09 \pm 0.68$ | OOM | OOM |

Table 14: Recall scores (%) on AML edge classification task. Best result is indicated with **bold**.

| Model | Small HI | Small LI | Medium HI | Medium LI | Large HI | Large LI |
|---|---|---|---|---|---|---|
| GIN | $50.06 \pm 2.62$ | $23.14 \pm 2.70$ | $55.67 \pm 5.77$ | $22.88 \pm 2.72$ | $52.68 \pm 0.32$ | $2.72 \pm 2.02$ |
| PNA | $59.65 \pm 1.82$ | $22.94 \pm 1.77$ | $64.79 \pm 3.07$ | $23.73 \pm 2.58$ | $63.19 \pm 0.74$ | $16.85 \pm 1.93$ |
| GenAgg | $57.52 \pm 2.59$ | $22.37 \pm 3.18$ | $60.06 \pm 3.42$ | $19.92 \pm 4.00$ | $54.86 \pm 2.92$ | $5.37 \pm 1.91$ |
| RGCNE | $55.10 \pm 2.91$ | $25.69 \pm 1.17$ | $58.29 \pm 0.81$ | $26.52 \pm 1.07$ | $48.91 \pm 2.34$ | $16.57 \pm 0.90$ |
| Multi-FraudGT | $\mathbf{72.02 \pm 0.96}$ | $34.42 \pm 1.16$ | $71.40 \pm 0.64$ | $32.14 \pm 0.82$ | $68.05 \pm 1.42$ | $24.70 \pm 0.43$ |
| Multi-GIN | $64.49 \pm 2.47$ | $31.37 \pm 2.46$ | $66.03 \pm 2.01$ | $28.47 \pm 3.36$ | $67.06 \pm 1.42$ | $6.83 \pm 7.25$ |
| Multi-PNA | $69.06 \pm 1.51$ | $29.95 \pm 1.86$ | $68.12 \pm 2.33$ | $32.55 \pm 0.67$ | $70.45 \pm 0.64$ | $21.94 \pm 2.11$ |
| Multi-GenAgg | $65.32 \pm 2.71$ | $29.55 \pm 1.95$ | $65.56 \pm 2.84$ | $30.94 \pm 0.41$ | OOM | OOM |
| MEGA-RGCNE | $66.83 \pm 0.96$ | $32.14 \pm 0.97$ | $68.09 \pm 1.37$ | $28.48 \pm 0.76$ | $68.26 \pm 1.42$ | $17.09 \pm 0.76$ |
| MEGA-GIN | $71.74 \pm 1.64$ | $33.25 \pm 0.87$ | $67.77 \pm 1.40$ | $28.18 \pm 3.52$ | $63.64 \pm 3.15$ | $6.91 \pm 1.13$ |
| MEGA-PNA | $71.48 \pm 1.32$ | $\mathbf{35.89 \pm 0.65}$ | $\mathbf{73.07 \pm 0.39}$ | $\mathbf{36.69 \pm 0.72}$ | $\mathbf{71.15 \pm 0.43}$ | $\mathbf{29.10 \pm 0.74}$ |
| MEGA-GenAgg | $71.14 \pm 1.72$ | $35.62 \pm 0.50$ | $69.53 \pm 1.40$ | $32.60 \pm 0.18$ | OOM | OOM |

offer deeper insights into the performance characteristics of our proposed methods across varying data regimes.

### E.1 ADDITIONAL COMPARISONS

Table 15: Comparison with DIAM on AML edge classification task.

| Model | Small HI | Small LI |
|---|---|---|
| DIAM | $51.82 \pm 6.09$ | $9.80 \pm 1.50$ |
| MEGA-PNA | $74.01 \pm 1.55$ | $46.32 \pm 2.07$ |

On the illicit transaction detection task, MEGA-PNA clearly outperforms DIAM (see Table 15), demonstrating the advantages of modeling edge-attributed multigraphs to capture complex transaction patterns. Unlike DIAM, which is a specialized solution for ETH node classification task Chen et al. (2021a), our model MEGA-GNN is a general-purpose architecture capable of both node and edge classification on multigraphs. It generates expressive edge embeddings and propagates them through the message passing process.

## F BROADER IMPACT

This work contributes effective graph machine learning techniques for financial crime analysis by addressing the specific challenges posed by multigraph structures in financial transaction networks.

Our model learns to detect illicit behavior directly from data, rather than relying on predefined, rule-based systems. This end-to-end approach improves adaptability and detection performance.

By enabling more accurate detection of illicit activity, our model has the potential to support financial institutions and regulatory bodies in identifying and preventing illicit financial behavior, such as money laundering and fraud. This may lead to stronger financial oversight, reduced criminal financing, and overall societal benefit through enhanced economic transparency and security.

## G    LLM USAGE

LLMs were used solely as a writing assistant to polish the language of this manuscript, such as checking grammar and improving clarity of expression. They were not used extensively.

