# OpenReview forum: "MEGA: Message Passing Neural Networks for Multigraphs with EdGe Attributes"
_ICLR.cc/2026/Conference — Submitted to ICLR 2026_

### Official Review · Reviewer_fwjv · 2025-10-26

**Soundness:** 2
**Presentation:** 3
**Contribution:** 2
**Rating:** 2
**Confidence:** 5

**Summary:**

This paper introduces MEGA-GNN, a graph neural network structure designed for multigraphs with edge attributes, addressing an important problem with potential applications across various domains. The paper is well written and easy to follow, and experiments are conducted on several real datasets for different tasks. However, the technical novelty of MEGA-GNN is limited, as its integration with existing GNN baselines and the two-stage aggregation approach are not fundamentally new. The architecture closely resembles standard GNNs, with only minor extensions for multiedge aggregation. Additionally, the experimental evaluation lacks depth, and the presentation of results is unclear, making it difficult to assess improvements over existing methods.

**Strengths:**

1. The paper addresses an important problem concerning multigraphs with edge attributes and proposes a graph neural network structure suitable for various applications.
2. The paper is well written and easy to follow.
3. The experiments are performed on several real-world datasets across different applications.

**Weaknesses:**

1.	At line 92, the authors state that MEGA-GNN can seamlessly integrate with GNN baselines; however, this may not be a significant advantage or contribution, as it suggests that the proposed MEGA-GNN has limited novelty or differences compared to existing GNNs. This actually highlights the limited technical contribution of this work.
2.	The two-stage aggregation in Definition 4 appears natural but lacks novelty, as it simply introduces another layer of aggregation over the single-stage aggregation. This idea is not new.
3.	In Section 3.3, apart from the aggregation over multiedges, the approach is quite similar to existing GNN architectures. Moreover, the aggregation design in Section 3.3 does not present significant challenges.
4.	The experiments are conducted on three quite different tasks, which makes them more general but insufficient in depth.
5.	There are many existing methods for ETH node classification that should be compared. For example:
-	Enhancing graph neural network-based fraud detectors against camouflaged fraudsters
-	Effective Illicit Account Detection on Large Cryptocurrency MultiGraphs, CIKM
-	BERT4ETH: A Pre-trained Transformer for Ethereum Fraud Detection. In Proceedings of the ACM Web Conference

I understand that these studies are focused on ETH anomaly detection, but fundamentally, their methods are designed for multi-graph neural network classification.

**Questions:**

1.	The presentation of the results in Figure 3 is confusing and difficult to understand. Why not provide a comprehensive table of results for the AML datasets? It is hard to determine whether your method improves upon existing methods.

---

> ### Author Response · Authors · 2025-11-20
> **Author Response 1/2**
>
> We thank the reviewer for carefully reading our work and providing detailed feedback. We appreciate their recognition that MEGA-GNN addresses an important problem concerning multigraphs with edge attributes. Below, we address the identified weaknesses and answer your questions:
>
> ### Weakness 1
> > We appreciate the reviewer’s comment; however, we respectfully disagree. MEGA-GNN’s ability to plug into diverse GNN backbones and consistently deliver substantial accuracy gains clearly indicates (1) its general applicability and (2) that the proposed two-stage aggregation is a genuinely novel and orthogonal modeling contribution not addressed by existing GNN baselines, including GIN, PNA, R-GCNE, and GenAgg.
>
> ### Weakness 2
> > Our work is the first, to the best of our knowledge, to identify and formalize the limitations of single-stage aggregation in the context of edge-attributed multigraphs, and to prove that a two-stage aggregation in message-passing strictly increases the expressive power of GNNs in this setting.
>
> > Crucially, MEGA-GNN is **not simply “adding another layer.”** As detailed in Figures 8 and 9 of Appendix C, we contribute a comprehensive and principled design. Our solution addresses the unique challenges of edge-attributed multigraphs, enabling key  capabilities that prior approaches for edge-attributed multigraphs lack (see Appendix A for further discussion). The consistent empirical improvements across diverse GNN baselines and datasets  demonstrate MEGA-GNN's effectiveness in practice.
> Although prior work has explored hierarchical or multi-level message aggregation, we acknowledge this in the Related Works section and clarify that these approaches do not extend to edge-attributed multigraphs.
>
> ### Weakness 3
> > As discussed in our previous responses, **MEGA-GNN** extends beyond standard GNN architectures by providing a **theoretically grounded** approach to aggregating information in edge-attributed multigraphs. Its **two-stage mechanism** consistently improves performance across a wide range of baselines. Overall, MEGA-GNN offers a principled and comprehensive solution for edge-attributed multigraphs, effectively addressing key limitations of prior methods. For detailed comparisons with existing approaches, please refer to **Appendix A**, and to explore the full complexity of our design, see **Figure 9 in Appendix C.**
>
> ### Weakness 4
> > We assess MEGA-GNN on 10 datasets, including seven financial transaction datasets (six AML and one Ethereum phishing detection) and three social interaction datasets. We use five different baselines (GIN, PNA, GenAgg, R-GCN, and FraudGT) and two other multi-graph adaptation methods (ADAMM and Multi-GNN) in our comparisons. In addition to the main results, we present deeper ablation studies. Table 2 shows the critical role of permutation equivariance, a property our method satisfies but Multi-GNN lacks. Table 3 quantifies the impact of our key design choices, notably, the two-stage aggregation combined with bi-directional message passing. Together, these analyses demonstrate the depth and the breadth of our evaluation as well as the  significance of MEGA-GNN's key theoretical and practical contributions.

---

> > ### Author Response · Authors · 2025-11-20
> > **Author Response 2/2**
> >
> > ### Weakness 5
> >
> > > We thank the reviewer for pointing out these relevant works. [1] primarily targets multi-relational graphs without edge attributes, which differs from our setting of edge-attributed multigraphs and is therefore not directly comparable. During the rebuttal period, we have  evaluated DIAM [2], the second paper you mentioned. We have not evaluated BERT4ETH [3], as DIAM appears to outperform it in the reported benchmarks and, given the limited time, we focused on the stronger baseline.
> >
> > > We will be happy to include the results for DIAM [2] on the ETH phishing account detection task in our revision. DIAM [2] uses the same ETH benchmark [4] that we use in our experiments (referred to as ETH-P in their experiments). In the following, we provide a comparison with DIAM [2] in our own experimental setting. The differences between our setup and theirs are;
> > > 1. We did not augment the data with "normal" accounts, instead we preserve the original illicit account ratio of 1/2552.
> > > 2. We perform a temporal split of accounts and transactions to create train/val/test graphs (see Appendix C.4).
> > > 3. We perform fully inductive training.
> >
> > **Inductive Setting on ETH Dataset as in MEGA**
> > | Model       |    F1               | PR-AUC          |
> > |-------------|------------------|------------------|
> > | **MEGA-PNA** | 64.84 ± 1.74     | 59.16 ± 1.58    |
> > | **DIAM**     | 57.28 ± 2.80     | 46.92 ± 5.58    |
> > | **DIAM$^*$** | 64.43 ± 1.07     | 54.86 ± 2.79     |
> >
> >
> > > In the table above, the **DIAM** results are obtained using the default hyperparameters from [2]. We also performed a hyperparameter search to ensure a fair comparison; the corresponding model is denoted as **DIAM$^*$**.
> >
> > > We observe that the F1-scores achieved by Multi-PNA, **DIAM$^*$**, and MEGA-PNA on this benchmark converge to roughly the same value, indicating that current methods appear to plateau around 64–65%.
> >
> > > Note that DIAM model and implementation does not support edge classification tasks, so we have not been able to evaluate it on our remaining benchmarks.
> >
> > - [1] Enhancing graph neural network-based fraud detectors against camouflaged fraudsters
> > - [2] Effective Illicit Account Detection on Large Cryptocurrency MultiGraphs, CIKM
> > - [3] BERT4ETH: A Pre-trained Transformer for Ethereum Fraud Detection. In Proceedings of the ACM Web Conference
> > - [4] Chen, Liang, et al. "Phishing scams detection in ethereum transaction network." ACM Transactions on Internet Technology, 2020.
> >
> >
> > ### Question 1
> > > As stated in line 404 of the paper, the complete tabular results corresponding to Figure 3 are provided in Appendix E. Figure 3 was intentionally designed to present the results in a more interpretable and comparative form than raw numerical tables.
> >
> > > Specifically, it visualizes the minority-class F1 scores for four baseline architectures (GIN, PNA, GenAgg, R-GCNE) combined with multigraph adaptations (Multi and MEGA) on AML benchmark. The upward trends from Base → Multi → MEGA across all architectures clearly show that MEGA consistently improves performance. While the figure is information-dense, we believe it offers a concise visualization of performance trends, complementing the detailed numerical tables provided in Appendix E.

---

> ### Author Response · Authors · 2025-11-25
> **Kind reminder to consider our response**
>
> Dear Reviewer,
>
> We would like to kindly remind you to review our responses to your comments and questions. We believe we have carefully addressed each point, and we would be happy to provide any additional clarification or information that might be helpful.
>
> Thank you very much for your time and consideration.

---

### Official Review · Reviewer_zf4d · 2025-10-28

**Soundness:** 4
**Presentation:** 4
**Contribution:** 3
**Rating:** 8
**Confidence:** 4

**Summary:**

This paper introduces MEGA-GNN, a novel graph neural network designed to address the challenges of modeling multi-edge interactions in graphs. The method utilizes a two-stage aggregation process, is designed to be permutation equivariant or invariant, and is evaluated on seven real-world datasets, showing practical application in areas like transaction networks.

**Strengths:**

- The paper tackles a practical and important problem (multi-edge graphs) that is common in real-world scenarios, such as transaction networks, but often overlooked in GNN research.
- The proposed method is proven to be a universal approximator. It is also shown to be capable of distinguishing between edges originating from the same neighbor versus those from different neighbors, a key capability for multi-edge graphs.
- The core improvement—the two-stage aggregation—is relatively simple to understand yet demonstrates effective results in the experiments.

**Weaknesses:**

- A significant clarification is needed regarding the use of timestamps. Timestamps are inherently ordered, which seems to conflict with the model's claim of permutation invariance.
    - How does the model handle this apparent contradiction?
    - Furthermore, does the model simply treat timestamps as static features, or does it properly account for the **temporal and causal relationships** implied by transaction data?
- The two-stage aggregation (separating edge-type aggregation) appears *prima facie* to be more computationally complex than a standard RGCN, which aggregates all edge types in a single step.
    - The paper lacks a clear computational complexity analysis. How does MEGA-GNN compare to RGCN in terms of time and memory?
    - Are there any efficiency benefits? If not, the authors should provide clear recommendations for practitioners on the trade-offs: when should one use MEGA-GNN (e.g., for high-dimensional edge attributes) versus a standard RGCN?

**Minor Comments**

1. The meaning of the '$+$' operator in '$m_A(x) + m_B(x)$' is ambiguous. Please explicitly define whether this represents element-wise addition, concatenation, or some other operation.
2. Figures 1 and 2 share many elements and convey very similar information. It would be clearer and more concise to merge them into a single, comprehensive figure.

**Questions:**

See above.

---

> ### Author Response · Authors · 2025-11-20
> **Author Response**
>
> We truly appreciate your positive assessment and the detailed, constructive feedback. We really appreciate your recognition of our work’s strengths and your helpful suggestions for improvement. Below, we address the identified weaknesses.
>
> ### Weakness 1
> > We appreciate the reviewer’s insightful observation. The inherent order of timestamps are only used to create temporal data splits (train/val/test) as described in Appendix C.4. Our model does **not** use this ordering in any part of its architecture. Within the model, timestamps are treated as regular static edge features, without imposing any sequential, causal, or temporal bias. In fact, the model operates identically even if timestamps are removed.
>
> > Permutation equivariance is therefore preserved: since the model never relies on an explicit ordering over nodes or edges, its outputs remain equivariant to any permutation of the nodes and edges in the graph (see Proposition 1).
>
> > The strict total ordering assumption appears only in the *theoretical universality analysis*. In the proof of Theorem 2 (see Appendix B.4), we assume the existence of a strict total ordering over graph edges, induced by an edge-labeling function L($\cdot$). Crucially, L($\cdot$) depends only on the edge features and is independent of the graph topology. Using these labels, we show that MEGA-GNN can construct unique node ids. This assumption is fully compatible with permutation equivariance, as it does not require storing or accessing the nodes or edges of the graph in any particular order.
>
> > We refer the reviewer to our response to W1 of reviewer ttSF for additional details. We apologize if this was unclear and will further highlight this in the revision.
>
> ### Weakness 2
> >We would like to clarify that our experiments use R-GCNE (i.e., R-GCN with edge updates) rather than the original R-GCN model (see Appendix C.6 for details). The original R-GCN neither leverages edge features nor performs edge updates, which leads to poor performance on the AML datasets.
>
> >Regarding computational complexity, Theorem 3 shows that MEGA-GNN’s two-stage aggregation has the same asymptotic complexity as standard GNNs with edge updates, including R-GCNE. Empirical results for training time and peak memory usage on AML-Small-HI are provided in Table 9.
>
> >Conceptually, R-GCN and MEGA-GNN address orthogonal but complementary settings. R-GCN is designed for labeled multigraphs with predefined relation types, whereas MEGA-GNN is designed for edge-attributed multigraphs where edge features are high-dimensional. When both forms of information are present, the two approaches can be combined. This is precisely how we construct MEGA-R-GCNE (see Appendix A.1 and the results in Figure 3), where we use a subset of edge attributes (line 398, currency types) to define relation types.
>
> >For practitioners, MEGA-GNN is most suitable for edge-attributed multigraphs; R-GCN is preferable for labeled multigraphs without informative edge attributes; and MEGA-R-GCN is appropriate when graphs are both relational and edge-attributed, allowing the model to benefit from both forms of structure.
>
> > We thank the reviewer for opening up this discussion, these clarifications are indeed valuable, and we would be happy to incorporate them into the revision if the reviewer finds them useful.
>
> ### Minor Comment 1
> > We thank the reviewer for raising this point. In our definition, the operator “$+$” in $m_A(x) + m_B(x)$ refers to **standard integer addition of multiplicities**. Each distinct element $x$, whether a scalar or a feature vector, is treated as an atomic item in the multiset. Therefore, when the same element appears in both multisets, we simply add their multiplicities as integers. We will make this explicit in the revised version to avoid any ambiguity.
>
> ### Minor Comment 2
> > We thank the reviewer for this suggestion. We had considered merging Figures 1 and 2 before the submission but ultimately decided to present them separately. Figure 1 provides a high-level visualization of single-stage versus two-stage aggregation, while Figure 2 is specifically designed to support the model equations introduced in Sections 3.3 and 3.4. The full architectural details are then provided in Figures 8 and 9 in Appendix C. This structure provides the reader with a top-down view of our approach, albeit presented in multiple steps.

---

> > ### Author Response · Authors · 2025-11-25
> > **Kind reminder to consider our response**
> >
> > Dear Reviewer,
> >
> > We would like to kindly remind you to review our responses to your comments and questions. We believe we have carefully addressed each point, and we would be happy to provide any additional clarification or information that might be helpful.
> >
> > Thank you very much for your time and consideration.

---

### Official Review · Reviewer_ttSF · 2025-11-05

**Soundness:** 3
**Presentation:** 3
**Contribution:** 3
**Rating:** 4
**Confidence:** 4

**Summary:**

The paper introduces MEGA, a message-passing framework for multigraphs that operates in two stages: first aggregating over multiple edges between the same pair of nodes, and then aggregating over different neighbors. This two-level design allows the model to distinguish intra-neighbor and inter-neighbor relations, which standard GNNs usually cannot. The authors also provide formal proofs showing that this two-stage mechanism is strictly more expressive than single-stage aggregation and maintains permutation equivariance.

**Strengths:**

1. The two-stage message passing mechanism is intuitive yet effective, and it directly addresses a structural limitation of current GNNs on multigraphs. The formal proof that this design is strictly more expressive than single-stage aggregation is quite convincing.

2. The paper carefully formalizes permutation equivariance and provides a universality result (under an edge ordering assumption). These theoretical discussions are coherent and help justify the architectural choices.

3. The experiments are thorough within their chosen domains, and MEGA achieves steady improvements over multiple strong baselines. The results are reproducible and the code organization appears clean and principled.

4. It’s nice to see that the authors didn’t just stop at the theoretical part, they also validated scalability and provided a reasonable complexity analysis, showing that the model can be trained efficiently in practice.

**Weaknesses:**

1. The universality argument depends on having a strict total order over edges. That’s fine for datasets with clear timestamps, but in reality many edges can share identical times or even lack ordering info. It’s not entirely clear how the model behaves in those cases, or whether the ordering assumption limits the practical generality.

2. There are some nice comparisons showing the effect of the two-stage design, but the analysis feels a bit narrow. For instance, it’s unclear how different aggregation choices or varying numbers of edge types would change the results. I would have liked to see a more systematic exploration here — even a small controlled study would help.

3. The datasets are mid-sized, and one of the baselines already runs out of memory. That makes me wonder how MEGA would perform on larger, real-world graphs. Some evidence or discussion about scaling behavior would make the contribution stronger.

4. Most results come from financial transaction or interaction graphs. That’s a relevant testbed, but I’m not sure how general these conclusions are beyond that context. It would be nice to see at least one example from another kind of multigraph, maybe something in social or transportation data.

5. The paper shows aggregate runtime comparisons, but not much insight into where the actual cost comes from. Since the model introduces an additional aggregation stage, some operator-level breakdown or profiling would make the trade-offs clearer.

**Questions:**

1. How does the model handle cases where multiple edges share the same timestamp or lack ordering information? Is there a deterministic tie-breaking rule, and does that affect permutation equivariance or expressiveness?

2. Could the authors provide more insight into how the two aggregation stages are parameterized? For example, is there any adaptive mechanism to select the number or type of aggregators based on edge multiplicity?

3. How expensive is the artificial-node construction in practice when using neighbor sampling? Does it cause additional indexing or memory overhead during batch training?

4. For the temporal datasets, how do you ensure there’s no information leakage across time? Are all multi-hop neighbors restricted to the same snapshot, or could future edges appear in the receptive field?

5. Have you considered pretraining or reweighting strategies to further mitigate the strong class imbalance in AML datasets? Some contrastive or cost-sensitive techniques might improve performance in the low-positive regime.

6. It would be interesting to see a plot showing how throughput and memory usage scale with the average number of edges per node. This would make the practical benefits of MEGA clearer to readers who care about deployment efficiency.

---

> ### Author Response · Authors · 2025-11-20
> **Author Response 1/2**
>
> We sincerely thank the reviewer for the detailed and thoughtful comments. We are pleased to hear that the strengths of our approach and analysis were clearly recognized. Below, we address the identified weaknesses and answer your questions:
>
> ### Weakness 1:
> > Thank you for your comment. First, we stress that the remaining theoretical contributions of our paper are completely independent of the strict total ordering assumption. Theorem 1, Proposition 1, and Theorem 3—respectively establishing the superior expressiveness of two-stage aggregation, the permutation equivariance of MEGA-GNN, and its inference complexity—do not rely on this assumption in any way.
>
> > **Our model can operate without any timestamps and any explicit ordering**. Universality is not required for, nor does it explain, MEGA-GNN's strong empirical performance. On the other hand, prior architectures such as Multi-GNN and Multi-FraudGT strongly depend on such an ordering to produce a consistent port numbering (see Appendix A). When such an ordering is not available, the accuracy of Multi-GNN and Multi-FraudGT drop significantly, whereas the accuracy of MEGA-GNN remains constant (see Table 2).
>
> > To avoid any ambiguity: the universality proof is included only to show that MEGA-GNN, like Multi-GNN, is universal when a strict total ordering is defined on the edges. MEGA-GNN itself does not require timestamps or any explicit edge ordering during its operation. We apologize if this was unclear and will further highlight this in the revision.
>
> ### Weakness 2:
> > Thank you for the comment. We interpret “different aggregation choices” as referring to the multi-edge aggregation used in our two-stage design. As discussed in Appendix C.5, MEGA-GNN variants differ in how this operation is instantiated. In our experiments, we use the same type of aggregation as the base GNN so that the statistics learned in both stages remain consistent. Our evaluation already covers a broad range of aggregation mechanisms. GIN provides a simple DeepSets-style sum aggregation. PNA combines several statistical aggregators (max, min, sum, std) with a learnable scaling mechanism. In PNA, we also experimented with higher-order moments such as skewness and kurtosis, which did not improve performance. GenAgg, by contrast, provides a fully learnable aggregation framework capable of modeling nearly all commonly used statistical aggregators. Given this broad expressive capacity, additional hand-designed aggregators and manual effort would be redundant. For these reasons, we concluded that no further exploration of alternative aggregation functions was necessary.
>
> ### Weakness 3:
> > The Large AML datasets used in our experiments contain **180 million edges**, and to the best of our knowledge, **they are the largest publicly available AML datasets**. All experiments were run on a single low-end GPU with 24 GB of memory. Under these constraints, we encountered out-of-memory errors only when using GenAgg as a baseline for the Large AML datasets. In the same setup, prior multigraph extensions (i.e., Multi-GenAgg) also resulted in out-of-memory errors.
> While these issues could be resolved simply by using more expensive hardware—such as an NVIDIA H200 with 141 GB of memory—or by distributing the model across several GPUs, we concluded that such escalation was unnecessary for the datasets used.
> **Notably**, MEGA-PNA consistently achieved high accuracy across the datasets and never produced any memory errors, even under strict hardware limitations.
>
> ### Weakness 4:
> > The three user–item interaction datasets used in Table 4—Wikipedia, MOOC, and Reddit—are indeed derived from social and collaborative platforms: users editing Wikipedia pages, students performing actions in a MOOC online course, and users posting within Reddit communities. In addition to these three datasets, our evaluation also includes seven datasets from the financial crime domain (six AML datasets and one Ethereum phishing detection dataset). By evaluating MEGA-GNN across **ten diverse datasets spanning social platforms and financial crime analysis**, we provide strong evidence of the broad applicability and robustness of our proposed MEGA-GNN framework.

---

> > ### Author Response · Authors · 2025-11-20
> > **Author Response 2/2**
> >
> > ### Weakness 5:
> > > Table 3 in the paper includes an ablation study on the impact of two-stage aggregation and bi-directional message passing. To further clarify runtime trade-offs, we additionally measured inference time over the test set. When moving from **GIN** to **MEGA-GIN (GIN with Two-stage Agg.)**, the only architectural change is the additional multi-edge aggregation. This adds a **24.7%** slowdown in inference, but yields a **50.4% improvement in F1**. Which clearly highlighting the value of the two-stage aggregation. Bi-directional message passing is more expensive, as expected, since each layer performs message passing over both original and reversed edges.
> >
> > **Ablation Results (Original Model Sizes)**
> > | **Ablation**|**AML Small HI (F1)**| **Inference time/epoch (seconds)** |
> > |-|-|-|
> > | GIN|46.50 ± 4.11|1.21 s|
> > | MEGA-GIN (Two-stage Agg.)| 69.98 ± 2.02|1.51 s|
> > | MEGA-GIN + Bi-dir. MP| **72.50 ± 3.26**| 6.77 s|
> > | MEGA-GIN + Ego IDs + Bi-dir. MP| 70.83 ± 2.18| 6.82 s|
> >
> > > When we match the number of parameters across models (~160k), the overhead of two-stage aggregation decreases to **10.9%**, indicating that the cost is modest once capacity is controlled.
> >
> > **Runtimes with Matched Parameter Counts (~160k)**
> > | **Model**| **Inference time/epoch (seconds)** |
> > |-|-|
> > | GIN|2.01 s|
> > | MEGA-GIN (Two-stage Agg.)| 2.23 s|
> > | MEGA-GIN + Bi-dir. MP| 6.77 s|
> >
> > ### Question 1
> > > The presence or lack of the ordering information has no impact on the expressiveness of the two-stage aggregation or permutation equivariance of MEGA-GNN (see our answers to W1 above). We order the transactions based on their timestamps only when creating the temporal data splits (see our answer to Q4 below).
> >
> > ### Question 2
> > > As noted in our response to W2, we leverage learnable aggregators such as PNA and GenAgg, which can automatically discover the most effective aggregation parameters and functions during training, eliminating the need for manual exploration.
> >
> > ### Question 3
> > > In MEGA-GNN, the so-called “artificial nodes” are intermediate variables created after the first aggregation stage to represent aggregated multi-edges. We visualize them as nodes to make the equations more complete and easier to follow in Sections 3.3 and 3.4, but **they do not correspond to a permanent expansion of the graph**. Conceptually, this is similar to computing latent variables or hidden states within GNN layers: they exist only temporarily to facilitate computation. Figures 8 and 9 in Appendix C.1. show when these intermediate representations are obtained. Since they are computed on-the-fly, they do not introduce significant indexing or memory overhead during batch training.
> >
> > ### Question 4
> > > We thank the reviewer for their question. Our training is fully inductive. We adopt a strict temporal splitting strategy, which ensures **no information leakage** across train, validation, and test splits (as explained in Appendix C.4).
> > During splitting, transactions are sorted by timestamp, and two cut-off points $t_1$ and $t_2$ define the train, validation, and test splits. To preserve historical context, we build graph snapshots at $t_1$, $t_2$, and $t_\text{max}$. The training graph includes only training transactions; the validation graph includes training + validation transactions but is evaluated only on validation transactions; and the test graph includes all transactions with evaluation restricted to test transactions. This setup mirrors real-world financial systems, where models must detect anomalies in new transactions using accumulated history.
> >
> > > Regarding receptive fields: for a seed edge in the training set, edges with timestamps **later than the seed but still within the training split** can appear in its multi-hop neighborhood. These edges are “future” relative to the seed edge, but since they are still part of the training split, their inclusion does not lead to any leakage. We do **not** include any edges from validation or test splits in training receptive fields.
> >
> > ### Question 5
> > > We used a reweighting strategy on the binary cross-entropy loss to address the strong class imbalance in the AML datasets. The minority class is assigned a higher weight, treated as a hyperparameter and varied between 3 and 7, as shown in Table 7.
> >
> > ### Question 6
> > > Thank you for the question. Even though our submission focuses on model expressivity and performance (e.g., in terms of minority-class f1 score), we have  also provided throughput and memory usage results for some of the datasets in Appendix D. In practice, an increase proportional to the average node degree should not be expected because we set constraints on the size of the neighbourhood sampled per seed node and per hop (see Appendix C.3).

---

> ### Author Response · Authors · 2025-11-25
> **Kind reminder to consider our response**
>
> Dear Reviewer,
>
> We would like to kindly remind you to review our responses to your comments and questions. We believe we have carefully addressed each point, and we would be happy to provide any additional clarification or information that might be helpful.
>
> Thank you very much for your time and consideration.

---

> > ### Comment · Reviewer_ttSF · 2025-11-28
> >
> > Thanks for the authors’ response. I will consider adjusting my score accordingly.

---

### Author Response · Authors · 2025-12-02
**Author Final Remarks**

Dear Area Chairs and Reviewers,

We sincerely thank you for the constructive feedback. Below we summarize the main points that emerged. We first highlight the core strengths of MEGA-GNN as recognized by the reviewers:

- **Relevance:** The approach directly addresses a key limitation in current GNNs concerning multigraphs. This capability is crucial for modeling real-world scenarios like transaction networks and is often overlooked in GNN research.
- **Methodology:** The two-stage message passing mechanism noted as highly effective because it allows the model to distinguish between edges originating from the same neighbor versus those from different neighbors, which is a necessary capability for robust multigraph processing.
- **Theoretical Grounding:** The architectural choices are supported by a coherent theoretical analysis.
- **Strong Empirical Results:** The method achieves consistent improvements over strong baselines. Reviewers also commended the reproducibility of the results and the clean, principled code organization.

In the rebuttal, we have addressed all reviewers' concerns. We emphasize that the core methodology, architecture, and claims remain unchanged. Our revisions focused on clarifying theoretical notation, improving readability, providing supplementary runtime analysis and comparisons with an additional baseline (DIAM).

### Theoretical Clarifications:
- **Universality and Permutation Equivariance:** We resolved a confusion regarding edge ordering. We clarified the paper to explicitly state that, in Theorem 2 and Corollary 1, we assume the existence of injective edge-labeling function, L:E→[M], which is essentially what we use in our Universality proof (Theorem 2 in the Appendix B.4). Namely, if an injective edge-labeling function exists, then the edges are inherently strictly totally ordered. Conversely, if the edges are strictly totally ordered, then an injective edge-labeling function exists. In practice, our model never relies on an explicit ordering over nodes or edges, ensuring its outputs remain equivariant to any permutation of the nodes and edges.
- **Use of Timestamps:** We clarified that timestamps in our model are treated strictly as static edge features. They do not impose any temporal ordering or sequential bias. The model's operation is, in fact, identical even when these timestamp features are completely removed.
### Additional Empirical Validation:
- **Additional Runtime Analysis:** Our paper already includes a runtime analysis in Tables 10 and 11, reporting (i) training time per epoch and (ii) inference throughput in transactions per second. In response to the reviewers’ request for further clarification, we have now added a finer-grained and complementary analysis for the models investigated in our ablation study (Table 3). This analysis (Table 5) shows that while our two-stage aggregation adds 24.7% slowdown in inference, it yields a **50.4% improvement in F1 score**. Furthermore, when models are evaluated with matched parameter counts, the overhead of our two-stage aggregation decreases to 10.9%. Detailed discussions accompanying these findings have been added to Section 4. We thank the reviewers for their comments, which enabled us to make these valuable additions.
- **Comparisons with DIAM:** Responding to reviewer `#fwjv`, we have incorporated additional discussions (in Appendix A) and empirical comparisons with a prior method (DIAM) specifically designed for fraud detection within cryptocurrency transactions (ETH). Table 1 shows that on the ETH benchmark, the F1-scores achieved by DIAM, Multi-PNA, and MEGA-PNA converge to roughly the same value (64%), indicating that both MEGA-PNA and Multi-PNA are competitive against this highly specialised baseline.
- The DIAM model computes only node embeddings and does not support edge classification. We extended this model by introducing an edge-classification head that uses both DIAM’s node embeddings and the original edge features. This modification allowed us to apply DIAM to the AML datasets as well. The results, given in Appendix E, show that  DIAM is significantly outperformed by our MEGA-PNA model on the AML task, reflecting the superiority of our model in capturing complex money laundering patterns.
### Presentation and Readability:
- **Architectural Diagram Relocation:** The architecture diagram for MEGA-GNN has been moved from Appendix C into Section 3 to make the model description more accessible.
- **Refined Figure 4:**. In the previous figure, the outer x-axis listed the GNN baselines, and the line-plot legends indicated the  datasets. In the revised version, we inverted this presentation: the outer x-axis now lists the datasets, while the line-plot legends denote GNN baselines, making the comparisons between baselines clearer. One reviewer pointed out the need for this, and we believe the updated figure is easier to interpret.

We thank the reviewers for their valuable comments and suggestions.

---

### Meta-Review · Area_Chair_zDdF · 2026-01-07

**Summary:**

This paper proposes MEGA, a two-stage message-passing framework for multigraphs that cleanly separates intra-neighbor and inter-neighbor aggregation.

Reviewers appreciated the clear presentation and consistent empirical improvements on real-world datasets. However, concerns remain regarding the practical assumptions underlying the universality result (e.g., strict edge ordering), limited analysis of scalability and computational trade-offs, and the relatively narrow empirical scope. While the idea is well executed, the novelty and generality are limited.

**Reviewer Concerns:**

Reviewers have concerns about incremental novelty, limited analysis of scalability and computational trade-offs, and narrow empirical scope.

**Reviewer Scores:**

Reviewers did not change their scores after the rebuttal.

---

> ### Public Comment · ~Kubilay_Atasu1 · 2026-02-25
> **Clarifications on Key Points Raised in the Meta-Review**
>
> We sincerely thank the reviewers and the meta-reviewer for their time and thoughtful feedback. We would like to respectfully clarify several points that we believe may have been misunderstood during the review process:
>
> **1- Strict Edge Ordering**
>
> In our rebuttal, we clarified that our model and evaluation procedure do not rely on any explicit ordering over nodes or edges. As a result, the model outputs remain equivariant to arbitrary permutations of nodes and edges. We hope this resolves the concern regarding potential ordering dependencies.
>
> **2- Scalability**
>
> Regarding scalability, our rebuttal noted that the proposed models were trained and evaluated on the largest publicly available AML datasets (approximately 180 million edges) using a modest GPU setup. These experiments suggest that scalability is not a practical limitation of the approach. In addition, Theorem 3 formally establishes that the asymptotic inference complexity of our method is no higher than that of standard GNN architectures that support edge features and perform edge updates.
>
> **3 - Score Changes**
>
> During the discussion phase, one reviewer indicated a potential intention to raise their score. However, due to procedural constraints related to identity leakage, score modifications were not permitted. We fully understand and respect these policies, and mention this only for completeness of the record.
>
> **4 - Generality Across Domains**
>
> We aimed to demonstrate the generality of our approach by evaluating it across multiple benchmarks, including datasets from social and collaborative platforms. Furthermore, on financial datasets, our model matched or outperformed a more specialized baseline (DIAM), suggesting that the proposed framework is broadly applicable rather than domain-specific.
>
> **5 - Novelty**
>
> One reviewer raised concerns regarding novelty and suggested additional comparisons. We addressed these points in our rebuttal, noting that several of the suggested methods either do not support edge features or are not designed for edge classification (e.g., DIAM), which limits their suitability and performance when used directly as comparison points.

---

### Decision · Program_Chairs · 2026-01-26

Reject